# Putative Molecular Mechanisms Underpinning the Inverse Roles of Mitochondrial Respiration and Heme Function in Lung Cancer and Alzheimer’s Disease

**DOI:** 10.3390/biology13030185

**Published:** 2024-03-14

**Authors:** Atefeh Afsar, Li Zhang

**Affiliations:** Department of Biological Sciences, University of Texas at Dallas, Richardson, TX 75080, USA; atefeh.afsar@utdallas.edu

**Keywords:** Alzheimer’s disease, cancer, mitochondria, heme, heme oxygenase

## Abstract

**Simple Summary:**

Well-balanced mitochondrial respiration and function are crucial for human health. Altered mitochondrial function underlies the pathogenesis of both Alzheimer’s disease (AD) and cancer. Elevated mitochondrial respiration supports cancer development, while reduced levels contribute to AD. Lung cancer particularly depends on mitochondrial respiration. Heme, essential for mitochondrial respiration, influences both diseases. Shared biological mechanisms like Pin1, Wnt, or p53 signaling operate differently in cancer and AD. In cancer, they lead to excessive cell growth and survival, whereas in AD, they induce cell death and neurodegeneration. Moreover, common risk factors like aging, obesity, diabetes, and tobacco usage impact the development of both conditions.

**Abstract:**

Mitochondria are the powerhouse of the cell. Mitochondria serve as the major source of oxidative stress. Impaired mitochondria produce less adenosine triphosphate (ATP) but generate more reactive oxygen species (ROS), which could be a major factor in the oxidative imbalance observed in Alzheimer’s disease (AD). Well-balanced mitochondrial respiration is important for the proper functioning of cells and human health. Indeed, recent research has shown that elevated mitochondrial respiration underlies the development and therapy resistance of many types of cancer, whereas diminished mitochondrial respiration is linked to the pathogenesis of AD. Mitochondria govern several activities that are known to be changed in lung cancer, the largest cause of cancer-related mortality worldwide. Because of the significant dependence of lung cancer cells on mitochondrial respiration, numerous studies demonstrated that blocking mitochondrial activity is a potent strategy to treat lung cancer. Heme is a central factor in mitochondrial respiration/oxidative phosphorylation (OXPHOS), and its association with cancer is the subject of increased research in recent years. In neural cells, heme is a key component in mitochondrial respiration and the production of ATP. Here, we review the role of impaired heme metabolism in the etiology of AD. We discuss the numerous mitochondrial effects that may contribute to AD and cancer. In addition to emphasizing the significance of heme in the development of both AD and cancer, this review also identifies some possible biological connections between the development of the two diseases. This review explores shared biological mechanisms (Pin1, Wnt, and p53 signaling) in cancer and AD. In cancer, these mechanisms drive cell proliferation and tumorigenic functions, while in AD, they lead to cell death. Understanding these mechanisms may help advance treatments for both conditions. This review discusses precise information regarding common risk factors, such as aging, obesity, diabetes, and tobacco usage.

## 1. Introduction

Alzheimer’s disease (AD) and cancer have emerged as significant global public health challenges. Despite substantial advancements in both practical and theoretical medicine, the effective treatment and prevention of these conditions remain elusive [1]. AD is the most frequent neurodegenerative disease impacting a total of 24 million individuals worldwide [2,3]. Furthermore, cancer, another health issue, stands as one of the primary contributors to mortality. In 2020, approximately 19.3 million new cancer cases and 10 million cancer-related deaths were recorded worldwide [1,4]. Lung cancer stands as the leading factor for cancer-related deaths, causing almost 1.4 million fatalities annually [5].

Studies suggest a lower occurrence of AD in individuals with a history of cancer compared to a control group without cancer [6,7]. Disease etiology presents opposite biological pathways: cancer is characterized by the uncontrolled growth of tumor cells, while AD is linked to neuronal cell death [8]. Both groups of individuals with cancer and those with AD could be thoroughly investigated for genes upregulated in one set of conditions and downregulated in the other. Examples include Pin1, p53, and Wnt [9]. P53 is upregulated in neurodegenerative diseases but downregulated in cancer. Pin1 is primarily elevated in cancer but downregulated in AD [10]. WNT signaling is downregulated in AD [11]. Wnt is upregulated in various cancer types [12,13,14]. Pin1 is a unique protein that establishes a direct link between the pathophysiology of AD and several cancer types. The enzyme Pin1 illustrates potential mechanisms, being upregulated in numerous human cancers while downregulated in AD [6,15]. Moreover, p53, a tumor suppressor protein, becomes inactive in numerous cancer cells. Conversely, heightened p53 activity is associated with neurodegeneration in individuals affected by dementia [8]. The inverse relationship between AD and cancer suggests that susceptibility to one condition may protect the other [1]. One possible explanation for the inverse association between cancer and AD is that the individuals who live long enough to experience cancer may belong to a specific group of healthy individuals. The same factors that promote cancer survival may also protect against AD [6,9]. Another potential explanation to consider is survival bias. Developing cancer decreases the likelihood of living long enough to develop dementia [9].

Cancer and AD share some common risk factors including aging, obesity, diabetes, and tobacco [1,9,16,17,18]. Aging, possibly due to inflammaging, is associated with the development of chronic diseases, including neurodegenerative disorders, cancer, and type 2 diabetes (T2D) [19]. Approximately two-thirds of cancer cases occur after the age of 70. Likewise, advancing in age is a predominant risk factor for AD. After the age of 65, the risk of developing AD approximately doubles every five years [16]. Obesity, reduced physical activity, and a high-cholesterol diet are significant risk factors for both cancer and AD [1]. T2D is suggested to elevate the risk of AD and dementia by a factor ranging from 1.3 to 5.5 times [20]. In the USA, between 1988–1994 and 2010–2015, cancer mortality rates decreased but remained about 30% higher in adults with diabetes compared to those without [21]. Both type 1 and type 2 diabetes mellitus (DM) ultimately lead to abnormal insulin signaling [22]. Impaired insulin signaling exacerbates mitochondrial dysfunction and worsens oxidative stress [23]. Moreover, tobacco smoking induces mitochondrial oxidative stress. Cigarette smoke exposure triggers mitochondrial dysfunction through Sirt3 depletion, SOD2 hyperacetylation, and cardiolipin oxidation [24].

Mitochondria are vital cellular organelles that provide energy to support cell life. They also play a critical role in the process of cell death [25]. Some studies have reported mitochondrial dysfunction in AD and cancer [1,16,26,27,28]. The moderate generation of reactive oxygen species (ROS) by mitochondria in cancer cells contributes to cancer cell growth and proliferation [1,29]. Heme plays a pivotal role in oxidative metabolism and the production of adenosine triphosphate (ATP) through mitochondrial oxidative phosphorylation (OXPHOS) [30]. Heme metabolism has been linked to non-small cell lung cancer (NSCLC) [30,31,32,33]. Rates of oxygen consumption and heme biosynthesis were increased in NSCLC cells. Enhanced heme function and mitochondrial respiration contribute to the progression of lung cancer cells [34]. Mitochondrial flaws harm cells through heightened ROS production, leading to damage and eventual death. They also disrupt OXPHOS, resulting in energy depletion in AD [28]. Studies have reported dysregulated heme metabolism, and elevated levels of free heme (or iron) in the brain in AD [26,35,36]. The accumulation of iron in the brain might promote neurodegeneration through oxidative damage in AD [37,38]. In AD patients, excessive iron in the inferior temporal cortex may hasten cognitive decline [26,39]. Furthermore, heme binds to amyloid beta (Aβ), creating complexes with increased peroxidase and superoxide activities, which further exacerbate oxidative stress [26,35,36].

This review thoroughly explores how mitochondrial respiration and heme function affect both lung cancer and AD. It discusses common risk factors, including obesity, diabetes, tobacco use, and aging in both conditions. Moreover, it examines the specific roles of Pin1, Wnt, and p53 concerning these disorders. The aim is to deepen our understanding of both AD and cancer by combining these findings, thereby opening doors for further investigation. Ultimately, it aims to identify heme and mitochondria as potential therapeutic targets, which could lead to effective treatments for both AD and cancer.

## 2. The Role of Heme and Heme Oxygenase in Cancer and AD

Iron is the fourth richest element in Earth’s crust. It is also essential to cell survival because it is a part of the Heme molecule of haemoglobin and myoglobin, as well as the Fe–S cluster proteins [40]. Heme, a complex of iron with protoporphyrin IX, is essential for the function of all aerobic cells [41]. Heme is an essential prosthetic group or cofactor in many proteins and enzymes that are involved in the binding and usage of oxygen, such as nitric oxide synthases and cytochrome P450. Furthermore, heme is involved in the detoxification of ROS such as peroxidases and catalase. Mitochondrial heme is crucial for OXPHOS formation and function [30]. Heme metabolism has been associated with NSCLC [30,31,32,33]. Increased heme synthesis and uptake in NSCLC cells result in elevated mitochondrial heme levels and OXPHOS subunits. Consequently, this enhances oxygen consumption, ATP production, and tumorigenic capabilities in NSCLC cells [30]. According to Hooda et al. [34] cancer cells display elevated levels of the rate-limiting heme biosynthetic enzyme, 5-aminolevulinic acid synthase (ALAS), and heme uptake proteins HCP1 and HRG1, resulting in increased heme availability. This boosts the production of oxygen-utilizing hemoproteins such as cytochrome c, cytoglobin, Cox-2, and cytochrome P450. Consequently, oxygen consumption intensifies, and cellular energy is generated through respiration. This heightened cellular energy production, cell proliferation, migration, and colony formation. Considering the significance of elevated heme metabolism in NSCLC tumorigenesis, limiting heme availability could potentially represent an effective strategy to hinder lung tumor progression [31]. According to Sohoni et al. [30] Engineered heme-sequestering peptides (HSPs) decreased heme uptake and inhibited tumorigenic functions in NSCLC cells. HSP2 notably slowed the growth of NSCLC xenograft tumors in mice, accompanied by decreased oxygen consumption rates and ATP levels in the tumors. Moreover, reducing heme biosynthesis and uptake, such as lowering mitochondrial respiration, effectively decreased oxygen consumption, cancer cell migration, proliferation, and colony formation [34].

Aβ peptides, the primary components of plaques, are produced by sequential proteolytic cleavage of the amyloid precursor protein (APP) via β-secretase (BACE1) and the γ-secretase complex [42]. Levels of BACE1 and its activity are elevated in the brains and bodily fluids of individuals with AD [43]. Although BACE1 inhibitors reduce Aβ deposition, they do not improve cognitive function in patients, because of their impact on synaptic function [44]. Aβ and APP are present within mitochondrial membranes, interacting with mitochondrial proteins, thereby amplifying ROS generation. This process leads to structural and functional impairment within the mitochondria, disrupting normal neural function [26,45]. Multiple molecular mechanisms have been suggested in AD [42,46,47,48]. Heme metabolism has undergone alteration within the brain of AD patients [36,46]. Direct evidence of altered heme metabolism in AD brain was demonstrated by the increase in heme synthesis and heme levels [36,46]. AD patients and mice exhibit a decrease in the expression levels of the rate-limiting heme synthesis enzyme ALAS1 and heme degradation enzyme HO-2. Aβ lowers heme degradation and HO-2 levels, which are raised to support neuronal functions. Reduced heme metabolism, particularly reduced HO-2 and heme degradation levels, is probably a very early event in AD [49]. Aβ binds to both heme and heme-a to generate a complex known as Aβ-heme [36,46,48]. Heme-a may bind to Aβ with a higher affinity than heme-b, due to its hydrophobic farnesyl group, which is capable of binding with the hydrophobic regions of Aβ. This interaction could lead to an augmentation in absorbance within the visible spectrum range of the heme-a–Aβ complex [36]. Heme inhibited the aggregation of Aβ by forming Aβ-heme, indicating that Aβ-heme may impede Aβ aggregation in vivio. Moreover, the overproduction of Aβ in the AD brain may associate with and limit the bioavailability of regulatory heme, leading to a state of heme deficiency [36,46,48]. Heme deficiency in brain cells diminishes mitochondrial complex IV, triggers nitric oxide synthase activation, modifies APP, and disrupts iron and zinc homeostasis. The metabolic impacts of heme deficiency are identical to AD patients’ dysfunctional neurons [50]. Elevated heme synthesis might act as a compensatory mechanism in response to a decrease in free heme [36]. The activity and protein content of mitochondrial complex IV (cytochrome c oxidase) were reduced by 95% in heme deficiency. In heme-deficient circumstances, complexes I–III and catalase remained unchanged, while ferrochelatase was elevated. Notably, complex IV is the sole hemeprotein in the cell containing heme-a, potentially explaining its heightened susceptibility [51]. Aβ-heme complexes with heightened peroxidase and superoxide activities, intensifying oxidative stress [35,48]. Heme deficiency and heightened peroxidase activity from Aβ–heme establishes a vicious cycle amplifying oxidative damage. Heme deficiency induces an elevation in $H_{2}O_{2}$ production, which may act as a substrate for the peroxidase Aβ-heme [48]. The effects of heme on lung cancer and AD are summarized in Figure 1.

### 2.1. Ferroptosis in Cancer and AD

Ferroptosis is an iron-dependent form of nonapoptotic cell death [52]. Ferroptosis is characterized by the fatal formation of lipid peroxidation products catalyzed by iron [53]. Ferroptosis is gaining attention due to its role in disease pathology and the possibility that this mechanism could be activated to eliminate cancer cells [54,55]. According to Dixon et al. [54], there is no singular universally applicable ferroptosis pathway. Numerous distinct metabolites and proteins can independently initiate, promote, and regulate ferroptosis, without any being obligatory. Sensitivity to ferroptosis can be modified by various signaling and transcriptional networks within the cell. Ferroptosis inducers can bind to membrane pore protein 2 and membrane pore protein 3 on the outer membrane of mitochondria. This interaction alters the permeability of the mitochondrial membrane, lessens the sensitivity of channels to iron ions, restricts the outflow of substances from mitochondria, induces mitochondrial dysfunction, releases a substantial quantity of oxidizing substances, and ultimately causes ferroptosis [56,57]. Dihydroorotate dehydrogenase (DHODH) inactivation triggers significant mitochondrial lipid peroxidation and ferroptosis in cancer cells with low expression of glutathione peroxidase 4 (GPX4^*low*^). Additionally, it synergizes with ferroptosis inducers to amplify these effects in cancer cells with high expression of GPX4 (GPX4^*high*^). The DHODH inhibitor brequinar selectively suppresses GPX4^*low*^ tumor growth by inducing ferroptosis. Furthermore, combined treatment with sulfasalazine—a ferroptosis inducer— effectively suppresses GPX4^*high*^ tumor growth [58]. The mitochondrial tricarboxylic acid (TCA) cycle and electron transport chain promote cysteine-deprivation-induced ferroptosis by serving as the main source of cellular lipid peroxide production [59].

Ferroptosis has been implicated in a variety of neurodegenerative diseases, including AD [55,60,61,62,63,64]. The significant degeneration of motor neurons and subsequent paralysis induced by the ablation of Gpx4 indicates that the prevention of ferroptosis by GPX4 is imperative for the health and survival of motor neurons in vivio [61]. Through the imaging of cell fate conversion, Gascón et al. [62] observed that heightened oxidative stress hinders the effective direct reprogramming of neurons, leading to considerable cell death. They identify inhibitors of ferroptosis, antioxidants, and Bcl-2 as pivotal metabolic agents in enhancing the generation of induced neurons from a range of somatic cells and in vivo after brain injury. Hambright et al. [63] studied a mouse model with a conditional deletion in forebrain neurons of Gpx4, a key regulator of ferroptosis. The mouse exhibited pronounced deficits in spatial learning, memory function, and hippocampal neurodegeneration. The findings suggest ferroptosis plays a significant role in neurodegenerative diseases such as AD.

### 2.2. P53, Ferroptosis, and Heme

The tumor suppressor p53 controls the expression of a wide range of proteins crucial for various cellular processes, such as apoptosis, cell cycle arrest, DNA repair, metabolism, and even autophagy and ferroptosis [65]. Studies have indicated that p53 plays a significant role in regulating ferroptotic responses [66,67,68]. While ferroptosis is primarily controlled by GPX4, p53 activation can modulate it without apparently affecting GPX4 function. P53 can indirectly activate the function of ALOX12 through the transcriptional repression of SLC7A11, leading to ALOX12-dependent ferroptosis upon ROS stress [66]. P53 restricts cystine uptake and sensitizes cells to ferroptosis, via repressing expression of *SLC7A11* expression, a pivotal element of the cystine/glutamate antiporter. Remarkably, the acetylation-defective mutant p533KR, which fails to induce cell-cycle arrest, senescence, and apoptosis, fully preserves its ability to regulate *SLC7A11* expression and trigger ferroptosis upon ROS-induced stress [67]. Regulation of *SLC7A11* gene expression and the response to ferroptosis is maintained by p533KR but is absent in p534KR98 [68].

It has been demonstrated that p53 interacts with heme [65,69,70,71]. P53’s interaction with heme was studied using various spectroscopic methods. Despite increased conformational flexibility in p53, its oligomeric state and zinc-binding ability remain unchanged. However, heme binding reduces its affinity to a specific DNA sequence, and the inhibitory effect on DNA binding is partially reversible [65]. Numerous epidemiological and experimental studies conducted over decades have demonstrated the association between iron excess, resulting from genetic factors or excessive dietary intake, and the development of various human cancers [69,72,73]. The stability of P53, a tumor suppressor protein, is directly regulated by heme. A positive correlation between iron and heme levels in vivo suggests that excess iron in cancer may sustain heme synthesis, thereby directly impacting P53 stability and function [32,69]. Shen et al.’ study [69] suggests that p53 is downregulated during iron excess. Heme binding with p53 interferes with p53’s DNA interactions, causing nuclear export and cytosolic degradation of p53. Furthermore, iron deprivation hinders tumor growth dependent on wild-type p53, indicating an association between iron/heme balance and p53 signaling regulation. Apoptosis is induced in diverse cancer cell types by dual inhibition of MDM2 and PPM1D by amplifying the p53 transcriptional program through the eIF2α-ATF4 pathway. PPM1D inhibition induces eIF2α phosphorylation, ATF4 accumulation, and enhanced p53-dependent transactivation upon MDM2 inhibition. HRI-dependent eIF2α phosphorylation and heme depletion are caused by dual inhibition of p53 repressors [74].

### 2.3. Heme Oxygenase in Cancer and AD

Mammals have three different isoforms of heme oxygenase (HO): HO-1, HO-2, and HO-3 [75]. Among these, HO-3 is rarely studied [76]. The reaction products of HO include carbon monoxide (CO), biliverdin, and ferrous iron (Fe^2+^) [75,77,78]. Heme catabolism and several physiological functions are linked to HO enzymes [79]. The majority of the CO in our bodies comes from heme metabolism, and it serves as a crucial signaling molecule [80].

Some studies have identified elevated levels of HO-1 expression in various malignant tumors, including lung cancer [80,81,82]. In human tumors, both primary and metastatic, HO-1 expression was higher compared to noncancerous tissue. Furthermore, increased HO-1 expression correlated with lower overall survival rates in patients with lung adenocarcinoma [81]. In colorectal cancer, HO-1 overexpression decreases cell proliferation, promotes cell cycle arrest and apoptosis, and reduces cell migration. These effects rely on the presence of wild-type p53, as p53 knock-out HCT116 and p53-mutated HT29 colorectal cancer cell lines did not exhibit these effects [83,84]. P53 knockout embryonic stem cells demonstrate heightened levels of the HO-1 protein in comparison to the wild-type cell line. There exists a p53-dependent negative modulation of HO-1 protein stability and that p53 null phenotype is correlated with an altered ROS homeostasis in embryonic stem cells [85]. In lung carcinogenesis, elevated HO-1 is mostly localized in the cytoplasm of tumor cells, while non-cancerous tissue shows nuclear localization. HO-1 exhibited a positive correlation with both tumor stage and lymph node metastasis, indicating its association with the progression of NSCLC [86]. Moreover, HO-1 interacts with signal peptide peptidase (SPP), leading to the cleavage of HO-1 at its transmembrane portion [84,87]. The expression level of SPP exhibited notably higher in breast, colon, and lung cancer tissues compared to their normal counterparts [88]. This elevation is associated with poorer overall survival rates [84,87,88]. SPP promotes tumor progression, at least in part, by facilitating the degradation of mTOR inhibitor FKBP8. Lowering SPP levels in these cancers’ cell lines reduces cell growth and migration/invasion abilities [88]. Independently from HO-1 enzymatic activity, SPP-mediated intramembrane cleavage of HO-1 enhances HO-1 nuclear localization and cancer progression [87]. Due to the intramembrane cleavage of HO-1 by SPP, a t-HO-1 form was produced and underwent nuclear translocation [84]. Moreover, nuclear HO-1 promotes tumor cell invasion and proliferation both in vitro and in vivo without requiring its enzymatic activity [87,89]. For nuclear HO-1-enhanced tumor cell growth, migration, and invasion in vitro, acetylation is necessary [89]. The expression of HO-1 may be negatively regulated by CO, either to halt further activity in the tumor microenvironment or by changing the phenotype of macrophages from M2 to M1, resulting in fewer HO-1 expressing cells within the tumor [90]. CO and biliverdin protect normal cells from transformation in the early phase of tumorigenesis. However, as cancer progresses, they promote the growth and survival of tumor cells [91]. CO treatment in the form of CO-releasing molecules- 2 (CORM-2), induced apoptosis in NSCLC cells. This effect was achieved by down-regulating the anti-apoptotic molecule Bcl-2 and up-regulating the pro-apoptotic molecule Bax, as well as subsequent apoptosis-related molecules caspase-3 and cyto-c [92,93]. In the A549 cell line of NSCLC, treatment with the HO-1 activity inhibitor VP13/47 led to decreased HO-1 expression. This resulted in notable declines in cell viability, proliferation and heightened apoptosis, mitochondrial dysfunction, and oxidative stress levels [94]. Conversely, HO-1 plays an anti-tumor role in certain cancers, including lung cancer [84,95]. HO-1 has distinct functions at various stages of tumor formation. Preceding tumor formation, it serves to eliminate aging and dead cells, inhibit tumors, and safeguard normal cells [80]. The overexpression of HO-1 in NSCLC NCI-H292 cells led to a decrease in their proliferation, migration, and angiogenic potential, as well as inhibition of tumor growth [95,96]. In non-cancerous cells, excessive production of iron resulting from chronic HO-1 overexpression can potentially contribute to intracellular toxicity and cell death (ferroptosis) [39]. HO-1 is encoded by the *HMOX1* gene with a molecular weight of 32kDa [97]. Cells that have high levels of *HMOX1* show reduced levels of intracellular ROS [95,98]. *HMOX1* attenuates cell proliferation and metastasis, with notable effects on miRNAs. In vitro and in vivo data indicate that the interplay between *HMOX1* and miR-378 significantly modulates NSCLC progression and angiogenesis, suggesting miR-378 as a new therapeutic target [95]. HO-1 catalyzes the degradation of heme in its oxidized form, hemin (Fe^3+^ protoporphyrin) [99]. Hemin induces a stress-inducible protein Sestrin2 (SESN2) through ROS, which serves as a protective mechanism suppressing oxidative stress [80,100]. When comparing NSCLC tissues to corresponding non-cancerous lung tissues, a significant decrease in SESN2 expression was observed [101]. Knockdown of SESN2 in lung cancer cells decreases both cancer cell survival and migration, while leading to overproduction of ROS by blocking the oxidative stress response. Additionally, numerous lung cancer expression datasets indicate a negative correlation between SESN2 expression and patient survival [102].

The role of HO-1 as a pivotal molecule in the nervous system’s reaction to damage is highly complex and not yet fully understood [103]. The beneficial and detrimental roles of HO-1 in the brains of individuals with AD are widely acknowledged [39,104,105,106,107]. HO-1 plays a neuroprotective role in animal models of AD by providing defense against oxidative damage [104,108,109]. HO-1 protects against A$\beta_{1-42}$-induced toxicity through the generation of CO in AD [110,111]. Both HO-1 induction and CO donors appear to be promising potential strategies for protecting AD’s degenerative effects on both neuronal and non-neuronal cell types in the central nervous system (CNS) [111]. CO improves memory impairments in mice with AD. Additionally, it inhibits the cleavage of APP by decreasing BACE1 expression through the upregulation of SIRT1 expression and the inhibition of NF-κB signaling. Consequently, CO controls Aβ levels and provides the molecular mechanisms contributing to the suppression of Aβ formation [42]. CO can protect neurons from apoptosis induced by oxidative stress. The protection is not rooted in the inhibition of apoptosis-associated K^+^ efflux. Rather, it stems from the inhibition of AMP-dependent protein kinase (AMPK) activation, a factor implicated in the harmful effects of Aβ [110]. CO and iron modulate plasmatic coagulation in AD [112]. At low levels and brief durations, HO metabolites may lack function. In moderate quantities and exposure durations, they offer advantages such as anti-inflammatory effects, reduced oxidative stress, and protection of the blood–neural barrier [39]. Prolonged overexpression of HO-1 presents drawbacks in AD [39,105,106,107,113].

Overexpression of HO-1 levels within the brain led to an increase in tau aggregation by inducing tau phosphorylation [105,106]. Prolonged HO-1 overexpression led to cognitive decline in transgenic mice, assessed by the water maze test. HO-1 impacts tauopathy via two pathways: Firstly, it promotes CDK5 expression by accumulating ROS, produced by HO-1 downstream products of iron in neuro2a cell lines and mouse brain. Secondly, HO-1 triggers tau truncation at D421 both in vivo and in vitro [105]. HO-1 concurrently co-expresses and induces the aggregation of Aβ42 and Aβ oligomers in the hippocampus area. It also modifies the morphology of the synapse, impairing the neural circuit. Overexpression of HO-1 potentially damages synaptic plasticity in the early stages of the disease, leading to AD-like pathology and cognitive abnormalities [107]. Prolonged overexpression of HO-1 within astrocytes results in abnormal iron buildup and impaired mitochondrial function in the brain, leading to reduced cognitive ability [39,113]. The Nrf2/HO-1 signaling pathway is activated by the microinjection of cocaine- and amphetamine-regulated transcript peptide into the rat’s hippocampus, which attenuates the oxidative stress damage [104,109]. When HO-1 levels are appropriately induced by Nrf2, shield the cells from iron-dependent toxicity [39]. HO-2 contributes to the maintenance of heme homeostasis [114]. It has demonstrated neuroprotective effects both in vivo and in vitro. By reducing the levels of pro-inflammatory cytokines such as IL-1β, TNF-α, and IL-6 in macrophages, HO-2 can inhibit inflammatory pathways [79,108,115]. Table 1 summarizes the effects of heme oxygenase on AD and cancer.

## 3. Common Risk Factors in Cancer and AD

Risk factors for both cancer and AD include aging, obesity, diabetes, and tobacco use [1,9,16,17,18]. As individuals age, crucial intracellular processes governing cell survival, growth, and function become disrupted [1]. Obesity and T2D can harm the brain, increasing the risk of cognitive decline and AD [116]. Increased aerobic glycolysis promotes cell proliferation, potentially increasing the likelihood of cancer development. Conversely, decreased glycolysis observed with aging hinders cell survival mechanisms and advances neurodegenerative processes [1,29].

### 3.1. Aging, Cancer, and AD: Unraveling the Connection

As individuals age, the functioning and development of the immune system are adversely affected [1]. The main contributors to aging include the organs’ inability to repair DNA damage caused by oxidative stress (non-programmed aging) and the telomere shortening due to repeated cell division (programmed aging). Both factors are observed in individuals with chronic obstructive pulmonary disease (COPD), an independent risk factor for lung carcinoma [117]. Telomeres are comprised of repeating DNA sequences at chromosome ends. They are bound by a protective protein complex called shelterin, which inhibits them from eliciting a DNA damage response (DDR) [118]. The telomere function is impaired in both cancer and aging [118,119]. Telomeres progressively shorten with age, both in vitro and in vivo. Telomeres shorten with each cell division [120,121]. Cells that express telomerase do not experience telomere shortening. The majority of cancer cells are telomerase-positive [120]. Although telomere shortening restricts a cell’s ability to proliferate, it is also linked to higher tumor initiation rates [122,123]. In vivo research has demonstrated that transient telomere dysfunction in early or late stages of cancer development promotes chromosomal instability and carcinogenesis in telomerase-proficient mice [123]. In experiments conducted on zebrafish, shorter telomeres led to more frequent, faster-growing, and more invasive tumors. Additionally, shortened telomeres increased senescence and systemic inflammation. Zebrafish larvae with very short telomeres exhibited increased melanoma dissemination, indicating that telomere shortening, similar to human aging, promotes a chronic inflammatory environment that elevates cancer risk [122]. A recent two-sample Mendelian randomization investigation reevaluated the impact of telomere length (TL) on the etiology of lung cancer, confirming that longer TL significantly increases the likelihood of lung cancer in the Asian population [124]. Senescent cells (SCs) alter their metabolic activity but remain viable and resilient to cell death [125,126]. Upon becoming senescent, cancer cells are taken out of the cell cycle, potentially inhibiting further cancer growth. Additionally, these SCs, through their senescence-associated secretory phenotype (SASP), can kill nearby cancer cells and attract immune cells that aid in the removal of more cancerous cells [127]. Prolonged inflammation may result in DNA damage, contributing to cancer development [1]. The generation and release of SASP factors, capable of inducing inflammation, serve as a powerful method for recruiting immune cells such as macrophages, natural killer (NK) cells, neutrophils, and T lymphocytes, which are responsible for removing them. However, SCs can also interact with immune cells to evade elimination [128,129]. Chemotherapy leads to an accumulation of SCs in both cancerous and normal tissues. paradoxically, SCs within tumors can induce tumor relapse, metastasis, and resistance to treatment, partly through SASP expression [130]. Cellular senescence and its SASP contribute to age-related diseases. Targeting SCs through removal, SASP modulation, or cellular reprogramming offers a promising therapy for conditions such as cancer and neurodegeneration disease [131].

During aging, vital intracellular mechanisms governing cell survival, proliferation, and function become dysregulated [1]. AD affects one in ten people over 65 and becomes more prevalent with age. Aging leads to deteriorating functioning of brain mitochondria, considered a significant early factor in aging [132]. Mitochondria likely influence the aging process by accumulating mutations in mitochondrial DNA (mtDNA) and by the net production of ROS [133]. Oxidative stress rises gradually in aging brains, leading to mtDNA mutations [134]. Oxidative stress has the potential to trigger tau phosphorylation and aggregation, as well as increase the construction and accumulation of Aβ [26,135]. The accumulation of mtDNA mutations disrupts OXPHOS and an imbalance in antioxidant enzyme expression results in excessive ROS production during aging [135]. Brain insulin receptors diminish with age, especially in AD [136,137]. Insulin and c-peptide concentration within the brain are correlated and diminish as individuals age, just like the densities of brain insulin receptors. In individuals with sporadic AD, brain insulin receptor (IR) densities are lower compared to middle-aged controls but higher compared to age-matched controls [137]. Aging is associated with diminished insulin activity in peripheral tissues and potentially within the brain. The decline in brain insulin action might contribute to age-related cognitive impairments, disruptions in metabolic balance, and an accelerated pace of aging [138,139]. In a study comparing intranasal insulin’s effect on cerebral blood flow (CBF) between younger (20–26 years, n = 8) and older (60–69 years, n = 11) adults, older participants showed increased perfusion in the occipital cortical brain region and thalamus with intranasal insulin compared to a placebo. However, total flow through major cerebropetal arteries remained unchanged for both age groups [139]. Additionally, glucose metabolism parameters from the oral glucose tolerance test (OGTT) are linked to reduced microstructural brain parenchymal homogeneity in “younger” older adults, but this link was less pronounced in older age groups. Overall, the correlation between reduced insulin action and brain homogeneity seemed to weaken with age, showing more significance in familial longevity [140]. Among older adults without T2D, lower microstructural brain integrity is associated with elevated insulin levels and decreased peripheral insulin function and sensitivity [138].

### 3.2. The Role of Obesity in Cancer and AD

Although cancer and AD are associated with aging, their simultaneous occurrence in patients, regardless of age, is rare. The cause behind this unexpected clinical observation remains to be clarified, but obesity-related mechanisms could open new avenues for preventing and treating these diseases [1,141]. Fat cells generate numerous active substances, including leptin and adiponectin. Leptin promotes cancer growth while hindering AD development, whereas adiponectin can inhibit cancer progression but may advance AD [141]. Leptin plays a significant role in controlling both appetite and the body’s energy metabolism [142].

#### 3.2.1. Leptin in Cancer

Leptin levels among cancer patients can vary depending on canctypes and locations. For instance, elevated leptin levels correlate with breast, and lung cancers, while lower levels of leptin are associated with pancreatic cancer. However, the relationship between leptin levels and cancer risk remains ambiguous [143,144,145,146]. Several studies have investigated the role of leptin in cases of NSCLC [147,148,149,150]. According to Song et al. [147], leptin levels were markedly higher in lung cancer patients in both serum and tissue samples compared to controls. Notably, leptin showed a strong correlation with gender but not with other tumor-related factors. Moreover, serum leptin levels were significantly higher in NSCLC adenocarcinoma patients compared to those with squamous cell carcinoma [148]. Conversely, other studies reported reduced serum leptin levels in lung cancer patients compared to the control group [151,152]. Additionally, some investigations found similar leptin concentrations between cancer and control groups [153]. Furthermore, research involving 66 NSCLC cases and 132 healthy controls revealed that increased serum leptin levels were an independent risk factor for NSCLC, regardless of central obesity [149]. Although Du et al. [150] suggested leptin’s potential as a biomarker for NSCLC screening, its diagnostic performance is inconvincible compared to combined detection with more effective biomarkers. Moreover, in a meta-analysis, Tong et al. [152] discovered lower serum leptin levels in the weight-loss group compared to the group with sustained weight. However, leptin might not seem to play a significant role in the development of cancer cachexia. More thorough and consistent studies are suggested in the future to verify these results.

The leptin receptor (LepR) exists throughout the immune system, with studies demonstrating its involvement in regulating both innate and adaptive immune responses [154,155]. T cells, one component of the adaptive immune system, are created in the thymus. Upon encountering specific antigens, they activate and secrete growth cytokines. Regulatory T cells (Tregs), a subset of T cells, have immune suppressive function [108,156,157,158,159]. Activated effector T cells migrate towards and infiltrate the tumor site, identifying and attaching to cancer cells via the interaction between their T cell receptor (TCR) and their cognate antigen bound to major histocompatibility class I (MHCI). Subsequently, they eliminate the targeted cancer cells [160]. Observations have shown that leptin triggers the activation and proliferation of T cells [145,161]. Leptin also modulates CD4^+^ T cells activation toward a Th1 phenotype by stimulating the synthesis of IL-2 and IFN-γ, indicating its role in modulating a Th1 cytokine-production profile in these cells [161]. Additionally, leptin promotes the transition towards a pro-inflammatory type 1 T helper cells, secreting IFNγ, rather than an anti-inflammatory type 2 T helper cells phenotype, secreting IL-4. Additionally, it also encourages Th17 responses [155]. CD4^+^CD25^+^ T cells within lung tumors play a role in suppressing the host’s immune response, potentially contributing to lung cancer progression [162]. Moreover, leptin seems to decrease the development of Tregs, which are associated with poorer outcomes and lower survival rates in various cancers, including lung cancer [154,155,162]. Leptin also enhances the cytotoxicity of NK cells [154,155], which develop strong cytolytic activity against tumors [154,163].

#### 3.2.2. Leptin in AD

Controversial evidence exists regarding how leptin levels are affected in AD [164]. Some studies indicate reduced levels of leptin in both cerebrospinal fluid (CSF) and plasma among individuals with AD [164,165,166], while others report elevated levels of leptin [142,164,167]. Additionally, some studies show unaffected leptin levels in CSF and cerebral tissue [164,168]. These discrepancies could arise from various factors such as limited sample sizes, unconsidered confounding elements such as exercise or diet, and the potential misclassification of AD. Moreover, reliance solely on clinical criteria without confirmation from neuropathology or newer AD neuroimaging or CSF biomarkers may contribute to these conflicting findings [169]. These discrepancies might arise due to various factors, such as limited sample sizes, overlooked confounding elements such as exercise or diet, and the potential misclassification of AD. Reliance solely on clinical criteria without confirmation from neuropathology or advanced AD neuroimaging or CSF biomarkers could also contribute to these conflicting findings [169]. Studies have reported the disruption of leptin signaling in AD [142,164,170]. Aβ oligomers directly target the LepR, dampening leptin transmission through negative allosteric regulation after binding to LepR. This eventually affects the responsiveness of hypothalamic neurons to this hormone [170]. Leptin, through its signaling pathways, can alter the levels of Aβ by blocking β-secretase activity and increasing ApoE-dependent Aβ uptake. Furthermore, it can enhance Aβ clearance and degradation [171,172]. Research demonstrates leptin’s impact on neuroprotection [143,164,167,173,174], including various mechanisms such as the suppression of Aβ accumulation, removal of phosphorylated tau protein, protection against oxidative stress, and attenuation of apoptotic cell death [164,167]. Additionally, leptin’s effect on hyperphosphorylated tau protein levels is attributed to its inhibition of glycogen synthase kinase-3 beta (GSK-3β), reducing tau protein phosphorylation [165,172]. It modulates tau phosphorylation through pathways involving AMPK, Akt protein, and p38 protein [169,172,175]. Moreover, leptin prevents synaptic disruption [173]. Some studies in animal models and human research suggest a potential link between leptin and AD, due to its beneficial impact on the modulation of cognition and neuroprotection [143,174]. Leptin decreases Aβ levels via AMPK activation in vitro and alleviates memory loss in vivo AD models [175,176]. Together, the studies mentioned offer initial validation for the possible therapeutic uses of leptin signaling enhancement in AD brains [164]. In rat models, leptin treatment improves memory and long-term learning performance while also modulating hippocampal synaptic plasticity [167,177]. Blocking Aβ and LepR interaction could potentially improve both metabolic and cognitive impairments in AD [170]. However, despite the connections between leptin and AD, the data does not support leptin as an indicator for cognitive decline onset [178]. Plasma leptin levels did not predict cognitive decline or cortical thinning, irrespective of participants having Aβ (+) or Aβ (–) status [179].

#### 3.2.3. Adiponectin in Cancer

Adiponectin plays a crucial role in both energy metabolism and inflammation. Studies suggest a correlation between serum adiponectin levels and susceptibility to various cancer types, including lung cancer, in vivo [180]. Adiponectin receptors 1 and 2 directly affect tumor cells by binding and activating adiponectin receptors and downstream signaling pathways [181,182]. This may restrict cancer cell proliferation (i.e., activating AMPK). Additionally, it may directly impact tumor vessels by activating the caspase cascade and inhibiting the NF-κB pathway in endothelial cells, leading to endothelial cell apoptosis and suppression of tumor angiogenesis. It also boosts insulin sensitivity and has anti-inflammatory effects [182]. Several studies have investigated the role of adiponectin in cases of lung cancer [151,153,183,184,185]. While some studies noted no discernible difference in serum adiponectin levels between lung cancer patients and controls [185], others reported lower adiponectin concentrations in lung cancer cases [153]. Nevertheless, a single study found higher adiponectin levels among lung cancer patients in comparison to the control group [151]. NSCLC patients demonstrate a notable decrease in total adiponectin levels compared to healthy individuals, with a specific down-regulation of high molecular weight (HMW) oligomers. Furthermore, adiponectin expression is lower in lung adenocarcinoma compared to other subtypes, regardless of other factors [186]. Therefore, the precise role of adiponectin in lung cancer remains elusive [183]. Genetically increased circulating adiponectin offers protection against lung cancer but poses a potential risk for colorectal cancer [187]. Adiponectin hinders migration and invasion by reversing epithelial-mesenchymal transition in NSCLC carcinoma, presenting its promise as a therapeutic strategy for addressing NSCLC [188]. Furthermore, adiponectin decreases cell viability and duplication, while elevating cell apoptosis. Additionally, it induces heightened lipid peroxidation, assessed via TBARS assay, and simultaneously decreases nitric oxide release, both markers of cellular oxidative stress [180].

#### 3.2.4. Adiponectin in AD

Adiponectin crosses the blood-brain barrier (BBB) and reaches the brain parenchyma [189,190]. Studies relating adiponectin levels to AD are controversial [164,178,191,192,193,194,195]. While AD patients exhibited notably lower levels of CSF adiponectin compared to those with mild cognitive impairment (MCI) and normal controls. However, serum adiponectin levels were higher in both MCI and AD patients than in the control group [191]. Another study found AD patients had 33% higher serum adiponectin than MCI patients. While the levels of adiponectin in CSF remained similar between both groups, they exhibited a positive correlation with Aβ42 and cognitive function, particularly in women [192]. Conversely, both MCI and AD patients exhibited considerably lower serum adiponectin levels compared to controls [193]. However, no observed link exists between adiponectin levels and AD or vascular dementia, whether considering the entire group or analyzing men and women separately. The likelihood of dementia in individuals with high and low levels of HMW adiponectin was nearly the same [194]. Although HMW adiponectin may have a positive association with general cognitive functioning in women but not in men [195]. Adiponectin serum levels exhibited a significant increase in sporadic AD subjects compared to controls, utilizing commercially available immuno-assay kit [166]. Although there is an association between adiponectin and AD, as well as AD-related disorders, the available data does not support the idea that adiponectin might serve as an indicator of cognitive decline development [178]. Significantly, only participants exhibiting Aβ (+) status demonstrated a substantial correlation between plasma adiponectin levels and cognitive as well as brain structural changes over time. In this Aβ (+) condition, higher plasma adiponectin levels predicted a faster cognitive decline [179]. Due to conflicting data, further research is imperative to determine the potential diagnostic and clinical significance of measuring adiponectin levels in both blood and CSF for MCI and AD [164]. The roles of adiponectin within the brain are remarkably diverse. Evidence indicates its involvement in fundamental processes of brain physiology, including the regulation of neuronal excitability, synaptic plasticity, neuroprotection, neurogenesis, and the modulation of glial cell activation [164,176,190,196]. An enriched environment induces anti-inflammatory responses in the brain by targeting the activation profile of microglia through an adiponectin-dependent manner [196]. Globular adiponectin (gApN) demonstrates direct anti-inflammatory effects on microglia in vivo by decreasing the synthesis of IL-1β, IL-6, and TNFα. Additionally, gApN inhibits nitrosative and oxidative stress caused by lipopolysaccharide in microglia. These anti-inflammatory and antioxidant effects of gApN on microglia are mediated through the AdipoR1/NF-κB signaling pathway [190,196]. Adiponectin-homolog osmotin has demonstrated improvements in AD-like neuropathological features, including Aβ production and aggregation, synaptic dysfunction, and cognitive deficits. Silencing AdipoR1 reversed osmotin’s benefits and exacerbated brain pathology in AD mice [164,176]. Aged mice lacking adiponectin exhibited impairments in spatial memory and learning. These mice developed AD pathologies, such as elevated Aβ42, tau phosphorylation, microgliosis, hippocampal atrophy, and astrogliosis, along with increased levels of IL-1β and TNFα. This suggests a potential contribution of adiponectin to neuronal and synaptic loss in AD [164,191,197]. The observed increase in serum Adiponectin levels in AD might reflect systemic and compensatory mechanisms against neurodegeneration [191]. Decreased levels of adiponectin are associated with deregulated cerebral insulin signaling and AD pathogenesis in aged individuals or those with T2D mellitus (T2DM), particularly among those with decreased CNS adiponectin levels [197,198]. AdipoR1 and AdipoR2, two high-affinity adiponectin receptors in the brain, may make up for low levels of adiponectin in the CSF [199]. The suppression of adiponectin receptor 1 may lead to metabolic disorders such as obesity and diabetes, which further exacerbate spatial learning deficits, memory impairments, and AD pathologies [199,200]. In obesity, decreased adiponectin levels fail to trigger 5’–AMP-activated protein kinase AMPK-mediated signaling. This leads to the phosphorylation of IR substrate 1 at serine residues, consequently blocking the insulin-degrading enzyme-mediated Aβ clearance. This inhibition results in increased Aβ accumulation in the brain [201].

### 3.3. Understanding the Link between Diabetes, Cancer, and AD


#### 3.3.1. Diabetes in Lung Cancer

T2D is an independent risk factor in the onset of several cancers [202,203,204,205]. Among lung cancer patients, those with DM showed decreased survival compared to non-diabetic counterparts [204,206]. A meta-analysis by Yi et al. [207] indicated a correlation between DM and a higher occurrence of lung cancer in women, although no such association was observed in men. The presence of preexisting diabetes is linked to the overall mortality risk in women diagnosed with lung cancer [206]. While preexisting T2DM might facilitate distant metastasis in small cell lung cancer (SCLC), it is the use of insulin therapy, not solely preexisting T2DM, that adversely impacts the prognosis of SCLC patients. These findings suggest that enhancing blood glucose control and reducing insulin analog use may be crucial for improving the long-term survival of individuals with diabetes and SCLC [203]. Lactate dehydrogenase (LDH), carcinoembryonic antigen (CEA), and C-reactive protein (CRP) are biomarkers in routine patient assessments. They may predict cancer in diabetic patients, aiding oncologists in terms of patient prognosis at the onset of treatment. Additionally, these markers can assist diabetes specialists in suspecting cancer, particularly when unexplained glycemic occur [205]. In geriatric patients with metastatic colorectal cancer, CEA, carbohydrate antigen 19-9 (CA 19-9), LDH, CRP, and neutrophil-to-lymphocyte ratio (NLR) are associated with overall survival [208]. Cancer patients with diabetes often receive less aggressive treatment and face a poorer prognosis when compared to those without diabetes [209]. Hyperinsulinemia, by activating the IR on tumor cells, might promote tumor growth and advancement [210]. In the context of lung cancer, hyperglycemia is thought to stimulate the growth and invasiveness of lung tumor cells [211,212]. Despite these findings, the relationship between T2DM and NSCLC prognosis remains unclear and contradictory, necessitating more research. Notably, individuals with NSCLC and T2DM have shown a tendency toward extended overall survival [202]. This could, in part, be credited to the use of the antidiabetic medication metformin [202,213,214]. Metformin’s impact on lung cancer risk and survival rates in T2DM patients is notably associated with reduced risk and enhanced survival in lung cancer cases [213].

The selective toxicity of metformin on cancer stem cells (CSCs) suggests that CSCs exhibit a reverse Warburg effect (i.e., the shift from OXPHOS to aerobic glycolysis), relying heavily on OXPHOS [215,216,217,218]. In a situation where there is a high OXPHOS profile, metformin’s inhibition of the mitochondrial respiratory chain disrupts metabolic requirements leading to the apoptosis of cancer cells [215,219]. Both metastatic and CSCs maintain elevated mitochondrial OXPHOS. It is conceivable that during a metastatic surge, benign cells augment the expression of OXPHOS subunits to promote greater ATP generation [220]. Unlike T2D, type 1 diabetes (T1D) does not exhibit the same heightened cancer risk [210]. If hyperinsulinemia is the key link between increased cancer risk and T2D, individuals with T1D, who receive less exogenously administered insulin, might exhibit a different cancer risk pattern [210,221]. T1D is linked to a modest overall excess cancer risk, with specific cancers differing from those associated with T2D [221,222]. Specifically, T1D is correlated with a higher likelihood of developing stomach, cervical, and endometrial cancers [210,221].

#### 3.3.2. Diabetes in AD

Several studies suggest that T2D is a considerable vascular risk factor and contributes to the development of AD [26,223,224,225,226,227,228]. The likelihood of developing AD is twice as high in individuals with T2DM compared to healthy individuals [26,224,227,229]. Insulin’s diminished impact on its target tissues is known as insulin resistance [136]. T2DM is associated with brain changes through mechanisms such as vascular inflammation, oxidative stress, impaired insulin transport, insulin resistance, reduced insulin transport across the BBB, and glial cells [116,136,223,226]. Moreover, T2DM exacerbates Aβ and tau pathology through aberrant insulin signaling, causing neurodegeneration [22]. Impaired insulin signaling, which correlates with reduced cerebral energy metabolism, renders neurons increasingly vulnerable to the adverse effects of ROS, exacerbating mitochondrial dysfunction and worsening oxidative stress [23]. Insulin crosses the BBB through an IR-specific, vesicle-mediated transport process in the brain endothelial cells (BECs). Factors such as high-fat diet (HFD) consumption, nitric oxide inhibition, and astrocyte stimulation can regulate insulin uptake and transcytosis in BECs [230]. Stanley et al. [228] reported changes in insulin signaling in the AD brain. In humans, a decrease in the ratio of insulin levels in CSF compared to serum is noticed in cases of whole-body insulin resistance [226,231]. In older individuals, cerebral insulin resistance may partly be due to impaired insulin transport into the CNS, affecting brain neuronal function [232]. Elevated blood insulin before early AD may contribute to both AD pathology and insulin resistance, although human data is limited. Another perspective suggests initial Aβ buildup causing neuronal insulin resistance, followed by secondary hyperinsulinemia, exacerbating AD progression [228]. Brain insulin resistance, coupled with oxidative stress and neuro-inflammation, promotes Aβ accumulation and toxicity. Furthermore, Aβ toxicity leads to brain insulin resistance [233]. Cognitive decline, especially concerning memory function, and peripheral metabolic dysfunctions are both correlated with brain insulin resistance [136]. The use of peripheral insulin administration for diabetes treatment can potentially cause hypoglycemia in non-diabetic individuals and may be ineffective due to impaired insulin transport across BBB. Therefore, research has concentrated on intranasal insulin administration, where insulin travels via bulk flow to the brain along olfactory nerve channels and trigeminal perivascular channels, bypassing the BBB [234,235]. Intranasal insulin has an impact on CNS indicators such as functional MRI, EEG, or MEG [136,234,235]. Administering intranasal insulin, both acutely and over 21 days, improves episodic memory in individuals diagnosed with MCI or AD [235,236], and regulates Aβ levels in the early stages of AD [237].

### 3.4. The Link between Tobacco and the Onset of Cancer and AD

#### 3.4.1. Tobacco in Lung Cancer

Tobacco use is regarded as one of the greatest health concerns worldwide [238]. Former or current tobacco use stands as the primary factor contributing to the onset of COPD, which is widely recognized as a risk factor for lung cancer. Exposure to cigarette smoke is a shared cause of both conditions and is responsible for nearly 90% of cases [239]. Those exposed to secondhand smoke face a 25% higher risk of developing lung cancer [238,240]. The adverse effects linked to e-cigarette use are associated with heightened oxidative stress. Human bronchial and pulmonary epithelial cells exposed to e-cigarettes experience oxidative stress, leading to inflammation, cytotoxicity, and increased endothelial cell permeability [238,241]. Smoking stands as the primary cause behind most cases of COPD and lung cancer. Cigarettes contain stable free radicals and reactive nitrogen oxygen species (RNOS) [117,242,243]. Aqueous cigarette tar produces hydroxyl radicals, resulting in oxidative DNA damage. Additionally, cigarette smoke, along with inhalable fibers and dust, amplifies the production of these damaging hydroxyl radicals [243]. RNOS damages DNA, hindering DNA repair and apoptosis. When lung damage occurs, RNOS disrupts the protective mechanisms, promoting cell proliferation and ultimately leading to cancer development [117,242]. Oxidative stress resulting from RNOS in COPD can trigger lung cancer through DNA damage mechanisms such as point mutations, single-strand breaks (SSBs), double-strand breaks (DSBs), and DNA cross-linking [117,238,244]. Serum samples from individuals with COPD exhibit elevated markers indicating inflammation, endothelial activation, and extracellular matrix remodeling compared to those with lung cancer. COPD presence may affect these circulating biomarkers levels, some of which are linked to prognosis [245]. The inflammatory system associated with COPD includes various immune cells such as neutrophils, alveolar macrophages, CD8^+^ T cytotoxic cells, and CD4^+^ T lymphocytes, activated and recruited through chemotactic mediators released upon exposure to cigarette smoke [246,247]. There is a rise in innate lymphoid cells (ILCs), notably ILC1 and ILC3 cells. These lymphocytes are likely to collaborate in sustaining the neutrophilic inflammation observed in the lungs of COPD patients. This may explain why inflammation persists even after smoking cessation [247,248]. The inflammatory conditions in COPD lung may impact lung cancer development by inducing epigenetic changes, including DNA methylation, miRNA expression, and histone acetylation [117,247]. Nicotine binding to brain nicotinic acetylcholine receptors, alters gene expression, receptor expression, and neurotransmitter levels, fostering dependence. Tobacco combustion produces carcinogens such as polycyclic aromatic hydrocarbons (PAH) and N-nitrosamines, causing DNA damage and cancer risk for years after use [249]. Cigarette smoke exposure in MRC5 cells results in telomere problems, potentially due to elevated oxidative stress, leading to senescence induction and SASP activation [250]. A significant intake of fruits and vegetables correlates with a decreased occurrence of COPD in current or former smokers, though this association is not observed in non-smokers [251]. Notably, the use of β-carotene supplements raises the likelihood of lung cancer among smokers, regardless of the tar or nicotine content in the cigarettes. Therefore, smokers are advised to refrain from β-carotene supplements [252].

#### 3.4.2. Tobacco in AD

Tobacco smoke, beyond nicotine, contains components capable of causing direct neurotoxic effects [253,254]. CSF indicators of neurodegeneration, neuroinflammation, and oxidation are associated with smoking and an increased risk of AD [255]. Former or current smoking increases the risk of developing AD, linking smoking to AD neuropathology in both humans and preclinical models. Smoking-related cerebral oxidative stress might contribute to AD pathogenesis and an increased likelihood of AD development [256]. Smoking could potentially affect neurodegeneration in cognitively healthy men, rather than primarily contributing to cerebrovascular burdens. This suggests that smoking could be a significant risk factor for AD development [257]. Exposure to secondhand smoke (SHS) exhibited a significant relationship with memory decline among individuals aged 55–64 years. For each additional year of SHS exposure in this age group, there was an additional 0.01 decline in memory test results [258]. Environmental tobacco smoke (ETS) exposure raises the risk of dementia and AD. Implementing smoking cessation in public places could potentially slow down the global dementia epidemic [259]. A history of smoking is associated with a more rapid decline in function and reduced volume of the entorhinal cortex in individuals with MCI [260]. Individuals identified with MCI typically advance to more severe stages of dementia, with the progression rate linked to their initial cognitive impairment level. Nearly all of these individuals exhibit the neuropathological features of AD. Hence, MCI generally represents early-stage AD [261]. Nicotine may impact the phosphorylation of tau [262,263]. Nicotine elevates phosphorylated tau protein levels in neurons with low levels of BAG2 This increase occurs due to stimulation of p38 and ERK1/2 kinase activity, accompanied by elevated expression of p38 and MAPKAPK2 expression. Conversely, when BAG2 levels are higher, they reverse nicotine’s effects toward a reduction in phosphorylated tau protein levels. This reversal may occur as a result of BAG2 inhibition of ERK1/2 through Hsp90 association, which is crucial for ERK1/2 function. Additionally, it may occur via BAG2-mediated degradation of phosphorylated tau as a result of BAG2 phosphorylation by p38/MAPKAPK2 [263]. Nicotine reduces the toxicity of Aβ via regulating the homeostasis of metals [262,264]. Elevated concentrations of iron, zinc, aluminum, and lead in the CSF of smokers could potentially indicate that cigarette smoking is accelerating the onset of cognitive decline [265]. Copper and zinc levels modulate Aβ aggregation and AD progression. Nicotine regulates metal balance by reducing ROS, down-regulating APP via copper homeostasis, and decreasing free intracellular copper ions via increased expression of copper chaperone for superoxide dismutase [264]. The aggregation mechanism of the Aβ40 peptide appears to be influenced by four metal ions and five aromatic compounds commonly present in cigarette smoke. Metal ions such as Pb(IV), affect the formation of Aβ dimers and trimers. Meanwhile, hydrocarbons such as toluene impact larger, more hydrophobic structures like tetramers. Certain metal ions and hydrocarbons seem to counteract each other’s effects. Notably, the uncharged and hydrophilic nicotine molecule does not directly impact Aβ or its aggregation process [262]. A decrease in smoking rates could likely lead to a decrease in future AD prevalence [256]. Prevention of AD involves managing modifiable risk factors in terms of treatable medical conditions and lifestyle choices such as diabetes, physical activity, sleep, diet, and use of tobacco [20].

## 4. Common Signaling Pathways in Both Cancer
and AD: P53, Wnt, Pin1

Biological mechanisms shared between these two conditions, such as Pin1, Wnt, or p53 signaling, function in opposite ways. In cancer, they result in uncontrolled cell proliferation and survival, while in AD they cause cell death and neurodegeneration [1,9]. The genes Pin1, P53, and Wnt show upregulation in one context and downregulation in the other [9]. In neurodegenerative diseases, p53 is upregulated, while in cancer, it is downregulated. Similarly, Pin1 exhibits higher levels primarily in cancer but lower levels in AD [10]. Additionally, WNT signaling is downregulated in AD [11]. On the contrary, Wnt shows upregulation in various cancer types [12,13,14]. Notably, the protein Pin1 serves a significant dual function in cancer and neurodegeneration. It regulates oncogenic signaling pathways, such as cyclin D1 and p53, and exhibits heightened expression in human cancers. Interestingly, when Pin1 is deleted in mice, it correlates with neurodegeneration similar to AD [15,266,267].

### 4.1. P53 in Lung Cancer

The most frequent lesions found in human cancers are p53 tumor-suppressor gene mutations [268]. Somatic p53 mutations are highly prevalent in various tumors colon, lung, pancreas, and ovarian endodermal tumors. Notably, the impact of a p53 mutation acquired through inheritance differs from a somatic mutation, even though the same mutant alleles are present in both inherited and somatic p53 mutations [269]. Understanding why an inherited TP53 mutation reproducibly leads to tumors in various tissues at different ages suggests that an initial or truncal TP53 mutation in certain tissue-specific stem cells might initiate a benign tumor. However, in other tissue-specific stem cells, TP53 mutations do not result in a phenotype or any impact upon cell division until additional mutations happen, which can takes a long time [270]. Mehta et al. [271] examined TP53 transcript variants in 10,310 human tumors spanning 32 types from The Cancer Genome Atlas (TCGA) data. TP53 was highly expressed in most tumors (99% above the median and 75% above the 75th percentile). They identified only two variants, FL/Δ40TP53$\alpha_{T1}$ and an uncharacterized variant (uc010cne.1), while other TP53 transcripts were not detected. Kazantseva et al. [272] reported that elevated Δ133p53β on a wild-type TP53 background might contribute to glioblastoma’s tumor-promoting pathways by contributing to the immunosuppressive and chemoresistant environment. Understanding how Δ133p53β becomes prevalent in cancer, possibly through hypoxia, is crucial for understanding disease progression. Elevating Δ133p53 levels aids normal human somatic cells to be reprogrammed to a pluripotent stem cell state, potentially offering a non- or less oncogenic and mutagenic method to enhance the reprogramming process [273]. Autocrined leptin is prevalent in most NSCLC tissues, potentially offering an additional prognostic factor for patients. Autocrined leptin seems to promote NSCLC cell growth by positively regulating the PI3K/AKT/mTOR signaling pathway while negatively regulating the P53 pathway [274]. The p53-deficient mouse could serve as a valuable model for investigating the involvement of the p53 tumor-suppressor gene in tumorigenesis [268].

### 4.2. P53 in AD

Under stressful conditions, p53 has the potential to disturb the regulation of mitochondrial function, contributing to abnormal neuronal conditions and the occurrence of some neurological disorders [275,276]. Aβ directly activates the tumor suppressor gene p53, resulting in p53-dependent apoptosis, while presenilin (PSEN) mutations have also been observed to initiate cell death dependent on p53 [266,267,277]. Oxidative DNA damage led to the nuclear translocation of Aβ42 and elevated p53 mRNA levels in guinea pig primary neurons. Increased p53 expression was observed in sporadic AD brains and transgenic mice carrying mutant AD genes. Degenerating neurons in both models exhibited accumulation of Aβ42 and p53, suggesting their involvement in AD-related neuronal loss. Therefore, the intracellular Aβ42/p53 pathway may be associated with neuronal loss in AD [277]. In AD, cellular stress activates p53 to address DNA damage and oxidative stress. However, microtubule network breakdown and tau oligomer pathology lead to p53 accumulation outside of the nucleus, causing dysfunction and impacting vital cell functions such as DNA repair and apoptosis [278]. Astrocytes are vital for assisting motor neurons in both health and disease. Rat astrocytes exhibit age-dependent senescence and a marked decline in their capacity to support motor neurons [279]. P53 isoforms Δ133p53 and p53β in astrocytes regulate their toxic and protective effects on neurons. The downregulation of Δ133p53 or overexpression of p53 in astrocytes enhances SASP and non-cell autonomous neurotoxicity in a neuron-astrocyte co-culture system. Additionally, reconstituted expression of Δ133p53 in neurotoxic astrocytes prevents SASP and transforms them into neuroprotective astrocytes [280,281,282]. Restoring expression of the endogenous p53 isoform, Δ133p53, shields astrocytes against radiation-induced senescence, supports DNA repair, and prevents astrocyte-mediated neuroinflammation [283]. As oxidative stress and chronic inflammation are prevalent in neurodegenerative disorders, astrocyte senescence could be a common underlying factor. Potential treatments might involve lowering SASP factor levels or adding young astrocytes into the motor neuron environment using neural progenitor cell transplants [279].

### 4.3. Wnt in Cancer and AD

Wnt proteins, originating from the Wnt gene family, play crucial roles in cell processes such as proliferation and differentiation [1,141,284,285]. Mutated Wnt genes and abnormal Wnt signaling are associated with various types of cancer, while Wnt signaling also contributes to the impact of Aβ in the brain [141]. When the Wnt pathway is upregulated, it enhances the susceptibility to tumor development [1]. Wnt signaling is downregulated in AD as a result of Aβ neurotoxicity [11]. Wnt signaling can be increased by leptin, which promotes cancer cell growth. It may also have a protective role in AD by reducing Aβ toxicity. Conversely, Wnt signaling is decreased by adiponectin, potentially inhibiting cancer growth and promoting the progression of AD [141].

#### 4.3.1. Wnt in Lung Cancer

In both cell lines and patient samples, the Wnt pathway signaling is elevated in cancer compared to normal breast tissue [12]. Wnt signaling is increased under hypoxia aides in the development of lung cancer [13]. Wnt pathway disruption in cancer cells reduces their dependence on aerobic glycolysis, partly due to the Wnt-controlled enzyme pyruvate dehydrogenase kinase (PDK1). PDK1 inhibits mitochondrial OXPHOS and maintains the glycolysis-dependent nature of tumor cells [286,287]. Signal transducers in the canonical Wnt pathway include Dsh, GSK-3β, adenomatous polyposis coli (APC), β-catenin, Axin, and T-cell factor (TCF)/lymphoid enhancement factor (LEF). Among these, β-catenin acts as a core molecule [285,288]. The dysregulation of Wnt/β-catenin signaling, critical for embryonic development and tissue homeostasis, leads to β-catenin accumulation in the nucleus and increased oncogene transcription. As a result, this contributes to the progression of cancers such as colon, liver, pancreas, and lung [288]. The co-receptors LRP5 and LRP6 (LRP5/6) and the Wnt receptor Frizzled directly interact with one another. Direct LRP5/6 binding to frizzled inhibits tumor metastasis that is frizzled-regulated [289]. The canonical Wnt/β-catenin signaling cascade initiates when Wnts bind to frizzled and LRP5, which, in turn, downregulates Glycogen synthase kinase 3 (GSK-3) activity [290,291]. LRP6, linked to several cancer progressions including human triple-negative breast cancer, NSC, and others, shares structural similarities with LRP5 [290]. The LRP6 rs6488507 polymorphism, combined with tobacco smoking, significantly elevates the likelihood of developing NSCLC [290,292]. Blocking Wnt signaling through LRP6 decreased cancer cell self-renewal ability and seed tumors in vivo [292,293]. APC, known beyond its role in the destruction complex, is essential for Axin’s rapid transition following Wnt stimulation and its association with LRP6/Arrow, a key early step in pathway activation. Axin phosphorylation in both Wnt-off and Wnt-on states requires the tumor suppressor APC [11,294]. Alterations in both Axin1 and Axin2 have been observed in various human cancers and cancer cell lines [285,295,296]. Axin1 downregulation is common in lung cancer. X-ray-induced inhibition of lung cancer cells may be mediated by increasing the expression of Axin1 through genomic DNA demethylation and histone acetylation [295]. Upregulation of Axin2 is found in most colorectal cancers [296]. In NSCLC, promoter methylation is linked to reduced Axin2 expression. This epigenetic silencing of Axin2 could lead to the nuclear accumulation of wild-type β-catenin [297,298]. Low Axin1 expression in lung cancer patients correlates with disease progression and a poor prognosis [295]. Axin2 functions as both a tumor suppressor and an oncogene [11]. Changes in Axin2 are associated with poor survival in patients with early-stage disease [298]. Individuals with elevated Axin2 expression may experience a longer overall survival period compared to those with low expression levels in lung cancer. The Axin2 rs2240308 C/T variant could potentially reduce the risk of both lung and prostate cancer, especially in Asian descendants and population-based studies [299]. Axin2 downregulation is associated with worse overall survival in breast cancer patients. Its rs11079571 and rs3923087 polymorphisms confer vulnerability to breast cancer [300]. Contrary to its expected role as a tumor suppressor, Axin2 acts as a potent promoter of carcinoma behavior by enhancing the activity of the transcriptional repressor Snail1, triggering functional epithelial-mesenchymal transition (EMT) program and driving metastatic activity [11,285,301]. Figure 2 provides a summary of Wnt signaling in lung cancer.

#### 4.3.2. Wnt in AD

Dysfunctional Wnt signaling is associated with several human diseases, including AD [1,302,303]. In the aging brain, Wnt signaling decreases [304,305]. This signaling holds significant importance at the synapse, essential for synaptic plasticity and maintenance in the mature brain [304]. In the context of AD, deregulated Wnt signaling may increase synaptic vulnerability. Synapses are affected directly by decreased Wnt signaling [304,306]. Reactivation of the Wnt pathway can reverse synapse loss and memory deficits [306]. Aβ-induced dysfunctional Wnt signaling is crucial in neuronal degeneration and synapse impairment in AD [303]. Moreover, activating Wnt signaling can prevent Aβ peptide aggregation formation, while Apolipoprotein E ϵ4, a primary risk factor for AD, prevents Wnt signaling [305]. Furthermore, certain genes associated with the Wnt signaling pathway are linked to AD pathology and cognitive function. These alterations align with the elevated activity of GSK-3β, which is observed in AD and implicated in other neurological disorders [302]. There is a suggested connection between Aβ-induced neurotoxicity and decreased Wnt signaling activity, with lower cytoplasmic levels of β-catenin. Notably, inhibiting GSK-3β, a pivotal Wnt pathway regulator, using lithium, serves as a protective mechanism for rat hippocampal neurons against Aβ damage [307,308]. A Constitutively active form of GSK-3β is involved in the abnormal tau phosphorylation and the development of neurofibrillary tangles (NFTs), along with reduced β-catenin levels in the hippocampus of individuals with AD [303]. Blocking GSK-3β promotes nuclear translocation of β-catenin, triggering Wnt signaling activation [284]. Dickkopf-related protein 1 (DKK1) expression is increased in the AD brain [302,304,309]. Inducing DKK1 hinders the Wnt’s inhibition of GSK3-β, promoting tau protein phosphorylation and NFTs creation in neurons [309]. The expression of DKK1, a negative modulator of Wnt signaling, induced by Aβ might heighten GSK-3β activity, subsequently increasing in tau hyperphosphorylation [1]. Consequently, silencing or neutralizing DKK1 can activate Wnt signaling and offer neuronal protection [1,309]. Interestingly, Dkk1 is necessary for Aβ-mediated synapse loss, as synapses are protected from Aβ insult when Dkk1 is inhibited [304]. Transgenic murine models expressing familial AD mutations exhibit a marked decrease in β-catenin translocation to the nucleus [303]. A notable reduction in β-catenin protein levels shows an inverse relationship with elevated GSK3-β tyrosine activating phosphorylation, in addition to downstream effects associated with disease progression and cognitive decline [302]. Wnt proteins, glycoproteins modified by palmitoylation, bind to a cell surface receptor complex consisting of frizzled and low-density lipoprotein receptor-related proteins 5/6 (LRP5/6) [310]. By interacting with LRP5/6 Wnt coreceptors, Dkk1 impairs Wnt proteins’ ability to bind to Frizzled and LRP5/6, hence inhibiting canonical Wnt signaling. In AD patients, higher GSK-3β activity and decreased cytoplasmic β-catenin levels result from Dkk1’s inhibition of Wnt signaling [304]. Overexpression of LRP5/6 in neuroblastoma SH-SY5Y activates Wnt signaling, and increases the expression of proliferation-related genes. LRP5/6 overexpression rescues cells from oxidative stress-induced cytotoxicity and cell death. LRP5/6 overexpression inhibits GSK-3β activity and decreases tau phosphorylation [311]. Mutations in the TREM2 gene elevate AD risk. TREM2 boosts microglial survival via activating the Wnt/β-catenin signaling pathway, offering a potential therapeutic target for AD [312]. Figure 3 provides a summary of Wnt signaling in AD.

### 4.4. Pin1 in Cancer and AD

Previous studies on Pin1 have concentrated on its twofold function concerning the advancement of cancer (Pin1 activation) and AD (Pin1 inactivation) [18,313,314]. Pin1 is frequently overexpressed in various human cancers, such as prostate and lung cancers [15,315,316,317,318]. It promotes uncontrolled cell division and malignant cell transformation in various cancer models [315]. Pin1 inhibition triggers apoptosis in cancer cells [15]. Through binding and isomerization, Pin1 modulates cell cycle progression by influencing various regulatory proteins. It promotes G1 checkpoint progression and S phase progression, along with regulating mitotic proteins [315]. Abnormal Pin1 activation disturbs the equilibrium between oncogenic and tumor-suppressing molecules, favoring oncogenesis. This imbalance could potentially be restored using Pin1 inhibitors [318]. Pin1 contributes to cancer development by upregulation of more than 50 oncogenes or proliferation-promoting proteins while downregulating more than 20 tumor suppressors and proliferation-inhibitory proteins [317]. Furthermore, Pin1 directly elevates various metabolic regulators such as β-catenin, HIF-1α, and c-Myc. Targeting this metabolic reprogramming process has proven effective in inhibiting cancer progression [319,320]. Recent advancements in creating Pin1 inhibitors through structure-based drug design and natural compounds, aiming to inhibit cancer activities [316]. Pin1’s oncogenic functions make it a promising target for cancer therapy, with inhibitors such as ATRA and KPT-6566 showing efficacy in vitro and in vivo [315]. However, it will take more research to identify when and in which patients targeting Pin1 would be therapeutically beneficial because the effects of Pin1-based therapies may vary depending on the specific type of cancer or its history of development, as well as the specifics of a person’s neurodegenerative disease or neural biochemistry [321].

Some studies show the involvement of Pin1 in AD [322,323,324,325,326,327,328,329]. Notably, a reduction in Pin1 activity is observed in AD [322,323]. This decreased activity might hinder the NMDA receptor-mediated turnover of Shank3 and PSD95 proteins, while also increasing NMDA receptor- and Aβ oligomer-mediated degradation of Shank3 and PSD95 proteins. This could contribute to synaptic loss during the progression of AD [323]. Pin1 can bind to the phosphorylated tau on Thr 231 (pT231 tau). When incubating the Pin1 protein in sections of both normal and AD brain tissue, robust binding of Pin1 was observed within the cytoplasm of neurons in the AD brain sections, but not in the healthy brain [324]. Hyperphosphorylated tau exists in both the physiological transform and the pathological cis form [325,326,327,328,329]. The trans isoform is functional and subject to dephosphorylation and degradation. In contrast, the non-functional cis hyperphosphorylated tau cannot be dephosphorylated and degraded, tending to aggregate and form tangles [313,325]. Pin1 can speed up the conversion of the cis isoform to the trans isoform of pT231-Tau and restore its normal functionality. Interestingly, cis pT231-Tau is notably elevated in the brains of individuals with AD, and there is a notable connection with NFTs and decreased Pin1 levels [325,326,327]. The emergence of cis pThr231-tau in neurons occurs early in MCI and accumulates in degenerating neurons as AD progresses. It localizes specifically in dystrophic neurites, contributing to memory decline [328]. An antibody targeting the initial driver of neurodegeneration, cis P-tau, hinders brain damage and tauopathy [330]. The interaction between Aβ and tau is thought to exacerbate AD progression [325,331,332]. Aβ formation causes hyperphosphorylation of tau [26,331]. Pin1 overexpression decreases Aβ secretion from cell cultures, while knockout of Pin1 increases Aβ secretion [333,334]. Pin1 binds to the phosphorylated Thr668-Pro motif of APP (pT668-APP) and catalyzes pT668-APP from cis to trans transformation. The cis pT668-APP isoform promotes the processing of Aβ. Pin1 could prevent Aβ processing by catalyzing from a cis to a trans isoform [325,333,334]. Figure 4 provides a summary of the effects of pin1, p53, and Wnt.

## 5. Mitochondria in Cancer and AD

Mitochondria, essential cellular organelles, supply the energy necessary to support cell life and play a crucial role in the cell death process [25]. Some studies documented altered mitochondrial function in both AD and cancer [1,16,26,27,28]. In cancer cells, the moderate generation of ROS by mitochondria contributes to the growth and proliferation of cancer cells [1,29]. Dysfunctional mitochondria contribute to oxidative stress and stimulate the activation of inflammasomes. The persisting impaired mitochondria may initiate the NLRP3 inflammasome pathway and are a source of oxidants [335]. Certain stimuli can cause chronic inflammation and carcinogenesis. For instance, the development of lung cancer is linked to COPD resulting from damage to the lungs due to smoking, inflammation, and consequent DNA damage [336,337]. Persistent inflammation can boost cancer growth and progression, whereas, in AD, it potentiates neuronal cell death and brain degeneration [1].

### 5.1. Mitochondrial Changes in Lung Cancer

Mitochondria interact with the endoplasmic reticulum (ER) at the level of membrane contact sites known as (mitochondria-associated membranes) MAMs. This interplay supports essential functions of the two organelles in controlling cell proliferation/death, cell metabolism, and Ca^2+^ homeostasis, in both normal and disease contexts, such as cancer [338,339,340,341]. HO-1, a 32kDa protein, predominantly localizes to the ER, but is also found in mitochondria, caveolae, and the nucleus [84,342,343]. Calcium homeostasis is vital for supporting cell survival. When there is a decrease in the flow of Ca^2+^ into the mitochondria, it blunts OXPHOS activity, resulting in a decrease in ATP generation [339].

Disruptions in the mitochondrial Ca^2+^/ROS homeostasis are associated with impaired respiration, mitochondrial fission, and mitophagy, which triggers cell death pathways such as autophagy, apoptosis, and paraptosis in prostate cancer cells [338,344]. During early apoptosis, an increase in C16-ceramide levels is observed, likely from the conversion of mitochondrial sphinganine and sphingomyelin. Sphingosine, lactosyl-ceramide, and glycosyl-ceramide levels remain stable. Ceramide production in mitochondria rises when MAM sphingomyelin levels drop. These sphingolipid changes happen at MAMs, mitochondria, and the ER during early apoptosis, a pathway avoided by cancer cells [339,345]. ER-mitochondria contact sites serve as a main platform for decoding danger signals, including variations in Ca^2+^ homeostasis, affected by oncogenes and oncosuppressors, ultimately influencing cancer development or progression [339,346]. Another critical tumor suppressor gene, such as p53, is also found in MAMs. In this scenario, p53 associates with the sarco/ER Ca^2+^ ATPase (SERCA) pump to fill Ca^2+^ stores in the ER. Antineoplastic treatment or an apoptotic signal favors Ca^2+^ flow from ER to mitochondria, which enable apoptosis [339,347]. Low levels of thioredoxin-related transmembrane protein 1 (TMX1) in cancer cells elevate ER Ca^2+^, faster cytosolic Ca^2+^ clearance, and reduced Ca^2+^ transfer to the mitochondria, reduce ER-mitochondria contact, shifts bioenergetics away from mitochondria, and speeds up tumor growth. TMX1, with its thioredoxin motif and palmitoylation to target the MAM, plays a crucial role in ER-mitochondria Ca^2+^ flux. It acts as a thiol-based tumor suppressor, enhancing mitochondrial ATP production and promoting apoptosis [348]. Combretastatin A-4 phosphate (CA4P) reduces peripheral NSCLC tumor vessel oxygenation, subsequently decreasing tumor core oxygenation and anoxia. CA4P also elevated levels of enzymes involved in heme biosynthesis, uptake, and degradation, as well as oxygen-utilizing hemoproteins. It did not diminish mitochondrial function in resistant tumor cells, implying the role of increased heme flux and function in tumor regrowth and resistance post-vascular disrupting agents (VDAs) treatment [349,350].

Fascin promotes lung cancer metastatic colonization by enhancing metabolic stress resistance and mitochondrial OXPHOS. Fascin, along with mitochondrial filamentous actin (mtF-actin), maintains the homeostasis of mtDNA to promote mitochondrial OXPHOS. Disrupting mtF-actin abrogates fascin-mediated lung cancer metastasis while restoring mitochondrial respiration can rescue metastasis. Targeting the altered actin cytoskeleton might rewire mitochondrial metabolism and prevent metastatic recurrence [339,351]. Various natural compounds can induce paraptosis in diverse tumor cell lines [338,352,353,354]. An α, β-unsaturated carbonyl compound of ginger, known as 6-Shogaol (6S), causes significant cytoplasmic vacuolation and leads to cell death in both breast cancer cells (MDA-MB-231) and NSCLC cells (A549) [352,353]. The application of plumbagin results in paraptosis in triple-negative breast cancer cells (MDA-MB-231), NSCLC cells (A549), and cervical cancer cells (HeLa) [352,354]. Plumbagin triggers paraptosis in cancer cells by covalently altering newly synthesized proteins and blocking the proteasomal degradation of unfolded proteins [354].

### 5.2. Mitochondrial Dysfunction in AD

A primary hallmark shared by both cancer and AD is mitochondrial dysfunction [16]. Mitochondria are the major source of oxidative stress. Defective mitochondria generate less ATP but produce more ROS, which may be a significant contributor to the oxidative imbalance observed in AD [26,355]. The emergence of ROS-induced mitochondrial dysfunction elevates Aβ generation, creating a vicious cycle between mitochondrial dysfunction and ROS, as well as, the harmful effects of Aβ [26]. The accumulation of Aβ peptide within the brain triggers the formation of NFTs, inflammatory reactions, heightened oxidative stress, and impaired mitochondrial function, which are the root causes of cell death and dementia [26,135]. In AD, mitochondrial defects lead to increased ROS production, resulting in cellular damage, eventual cell death, and disruption of OXPHOS, which depletes cellular energy [28]. Chronic inflammation seems to contribute to the development of age-related diseases, such as AD, and cancer [1,15,337,356,357]. Elevated ROS levels promote the transcription of proinflammatory genes and the production of cytokines such as IL-1, IL-6, and TNF-α, as well as chemokines, resulting in neuroinflammation [357]. Conversely, inflammatory responses stimulate microglia and astrocytes to produce high levels of ROS, suggesting that neuroinflammation could act as both a cause and a consequence of persistent oxidative stress [26,135].

Changes in MAMs implicate in the pathogenesis of AD [27,358,359,360]. Both β and γ-secretases are found in MAMs and harbor Aβ protein precursor processing activities. Enhanced accumulation of neutral lipids associated with Aβ production is reversed by inhibiting β- or γ-secretases. A proteomic method revealed interactions between Aβ protein precursor and its catabolites with essential proteins of MAMs regulating mitochondrial and ER functions [358]. Significant Aβ production occurs at mitochondria-ER contact sites, including the outer mitochondrial membrane and mitochondria-associated ER membranes. This heightened production may disrupt ER, mitochondrial, and mitochondria-ER contact site function, potentially serving as a key step in the neurodegeneration process in AD [361]. MAM dysregulation appears early in vivo based on the molecular alterations of MAM components observed in the cerebral cortex of 3-month-old APP/PS1 mice [27,360]. C99, the 99-aa C-terminal fragment of APP, is found in MAM along with endosomes, where it’s typically processed quickly by gamma-secretase. In AD cell models, unprocessed C99 accumulates in MAM, causing elevated sphingolipid turnover, and altered lipid composition in MAM and mitochondrial membranes. This disruption affects mitochondrial respiratory supercomplexes assembly and activity, contributing to AD’s bioenergetic deficits [27,359,362]. Mitochondrial dysfunction, though an early disturbance in AD pathogenesis, is not the primary driver [362]. There is a proposal that heightened levels of unprocessed C99, rather than the presence of Aβ, are involved in the mitochondrial dysfunction observed in AD [359]. Most cases of early-onset familial AD (FAD) experienced mutations in the PSEN encoding genes (PSEN1 and PSEN2) [363,364,365,366]. The rate of mitochondrial respiration was examined by assessing oxygen consumption in FAD and controlling fibroblasts. When compared to control fibroblasts, FAD fibroblasts exhibit considerably higher basal and maximal oxygen consumption rates (OCR) [363]. A study of the role of the PSEN homolog SEL-12 in *Caenorhabditis elegans* revealed that mutations in *sel-12* disrupt calcium homeostasis, leading to mitochondrial dysfunction. In SEL-12-deficient animals, calcium transfer from the ER to the mitochondria results in mitochondrial fragmentation and dysfunction. Mutants such as *sel-12(ar131)*, *sel-12(ty11)*, and *sel-12(ok2078)* exhibit 1.5, 2.9, and 2.1 times more ROS signal than wild-type animals, respectively [364]. Mutations in PSEN2 associated with FAD also disrupt autophagy through alterations in Ca^2+^ homeostasis [365]. In transgenic mouse models with a FAD-mutant PS2, enhanced ER–mitochondria juxtaposition was noticed, indicating enhanced mitochondrial Ca^2+^ uptake upon ER Ca^2+^ release [366]. Moreover, mutations in the *C. elegans* gene encoding a PSEN homolog, *sel-12*, results in mitochondrial metabolic issues contributing to neurodegeneration via oxidative stress. In *sel-12* mutants, elevated ER-mitochondrial Ca^2+^ signaling increases mitochondrial Ca^2+^ content, stimulating mitochondrial respiration and superoxide production [363]. Astrocytes in AD display characteristic signs of disease pathology, such as elevated production of Aβ, changes in cytokine release, and disrupted Ca^2+^ homeostasis. PSEN1 ΔE9 astrocytes exhibit significantly higher intracellular ROS levels compared to isogenic control cells [367]. Furthermore, individuals with the ϵ4 allele of apolipoprotein E (ApoE4) are more likely to AD development than those with ApoE3. In an astrocyte-conditioned media (ACM) model, evidence demonstrates a significant increase in ER-mitochondrial communication and MAM function. This increase is measured by the synthesis of phospholipids and cholesteryl esters in cells treated with ApoE4-containing ACM compared to those treated with ApoE3-containing ACM [368]. Mitochondria play a pivotal role in aging through the accumulation of mutations in mtDNA and the increased production of ROS [133]. Some studies have revealed the underlying molecular mechanisms and cellular effects of mitochondrial deficiencies, as well as the abnormalities of the mitochondria in AD [369,370]. Swerdlow and Khan [370] proposed “mitochondrial cascade hypothesis” comprehensively explains various aspects of late-onset, sporadic AD. In this model, the inherited genetic makeup of an individual’s electron transport chain sets basal rates of ROS production. This mechanism controls the rate at which acquired mitochondrial damage accumulates. Mitochondrial damage, resulting from oxidative DNA, RNA, lipid, and protein damage, amplifies ROS production and triggers three key events. These events include a reset response generating Aβ, a removal response eliminating compromised cells, and a replacement response attempting cell cycle re-entry.

Consistent features of mitochondrial dysfunction in AD involve damaged mitochondrial bioenergetics, heightened oxidative stress, and disrupted mitochondrial genome. The importance of these issues in triggering mitochondrial dysfunction may differ based on the individual biological, environmental, and genetic characteristics of each AD patient. Nonetheless, any of these abnormalities can result in the other two, worsening neuronal dysfunction and the process of neurodegeneration [26,27]. Mitochondrial bioenergetic deficits occur early in AD, preceding dysfunction in mitochondrial electron transfer chain (ETC) complexes and global metabolic failure. This suggests that AD may be a metabolic neurodegenerative disease, with Aβ-related effects observed across diverse organisms [371]. Oxidative stress can trigger tau phosphorylation, promote aggregation, and elevate the production and buildup of Aβ [26,135]. ATP deficits precede behavioral issues and Aβ aggregation in Aβ-expressing nematodes, indicating a potential independent or causative role for bioenergetic deficits. Dysfunctional ETC complexes I and IV follow the ATP drop, as observed in muscle-specific *C. elegans* strain overexpressing human Aβ. Global metabolic failure, which is evident in older Aβ nematodes, results from mitochondrial bioenergetic deficits and dysfunction in complexes I and IV. Notably, significant metabolic effects occurred in whole nematodes despite low Aβ in neurons [371]. The associative learning competency of the neuronal Aβ
*C. elegans* strain underwent evaluation, revealing a behavioral phenotype indicative of diminished cognitive function. These worms, designed as an AD model, exhibit altered behaviors involving complex neuron interactions and signaling due to Aβ expression. When stimulated with serotonin, they notably produce far fewer eggs [372]. *C. elegans* worms expressing Aβ peptides have been created to imitate the pathological features of AD [373,374]. Several strains of transgenic worms have been created to generate Aβ peptides. The production of Aβ 1-42 peptides in muscle cells is stimulated by elevating the temperature in the *C. elegans* strain GMC101 [373,375]. The *C. elegans* model expressing full-length Aβ1-42 can be utilized for screening potential therapeutics and investigating the toxic mechanisms of Aβ. PBT2, an AD therapeutic, provided rapid and significant protection against Aβ-induced toxicity in *C. elegans* and significantly improved cognition [375]. In *C. elegans*, human tau is extensively phosphorylated at disease-relevant sites and undergoes conformational changes resembling those observed in AD [376]. Sorrentino et al. [377] investigated the impact of nicotinamide riboside (NR) on neuroblastoma cells expressing human Aβ and, in line with data from the *C. elegans* model, they noted a significant decrease in intracellular Aβ deposits along with elevated levels of OXPHOS proteins and mitochondrial stress response transcripts.

### 5.3. Wnt, P53, Pin1, and Mitochondria

Some studies have illustrated the involvement of mitochondrial regulation of Wnt pathway [378,379,380]. Loss of mtDNA, increased glycolysis, and decreased OXPHOS are caused by mitochondrial transcription factor A (TFAM) deficiency. Elevated expression of the tricarboxylic acid-cycle metabolites α-ketoglutarate suppresses Wnt signaling and tumorigenesis [378,379]. According to Costa et al. [380], a decrease in mitochondrial ATP production results in the induction of ER stress, which in turn lowers canonical Wnt/β-catenin signaling both in vivo and in vitro. The decrease in mitochondrial ATP production via sublethal doses of various drugs results in reduced Ca^2+^ stores in the ER [378,380]. Wnt signaling inhibits Aβ oligomer-induced mitochondrial permeability transition pore, protecting hippocampal neurons from death. This suggests that Wnt activation could serve as a therapeutic target for individuals with AD by directly affecting the mitochondria [378,381]. Wnt-5a activates mitochondrial dynamics, causing acute fission and fusion in rat hippocampal neurons. This Wnt-5a/Ca^2+^ signaling pathway regulates the mitochondrial fission-fusion process in hippocampal neurons, offering insights into Wnt-related pathologies and neurodegenerative diseases linked with mitochondrial dysfunction [382]. P53 has been demonstrated to engage with mtDNA polymerase γ and is crucial for the maintenance of mtDNA integrity in response to oxidative injury [383,384]. The absence of p53 leads to a notable rise in mtDNA vulnerability to damage, resulting in a higher frequency of in vivo mtDNA mutations. This phenomenon can be reversed by stable transfection of wildtype p53 [384]. Pin1 enhances the mitochondria translocation of PGK1, where it triggers the activation of PDHK1 to facilitate PDH-inhibited activity of PDHK1 [319,385]. Mitochondrial PGK1, operating as a protein kinase, phosphorylates and activates PDHK1, thereby suppressing mitochondrial pyruvate metabolism and facilitating the Warburg effect [385]. Pin1 promotes the stress-induced localization of p53 to mitochondria, both in vitro and in vivo. Upon stress-induced phosphorylation of p53 on Ser46 by homeodomain interacting protein kinase 2, Pin1 stimulates its mitochondrial trafficking signal. This process is also induced by RITA, a molecule that activates p53. Pin1’s role is crucial for inducing mitochondrial apoptosis by this compound. These findings suggest potential implications for treating tumors expressing p53 [386].

## 6. Conclusions

Altered mitochondrial function is a critical factor at the intersection of lung cancer and AD. Mitochondrial respiration and OXPHOS play pivotal roles in energy production, with metastatic and CSCs maintaining elevated mitochondrial OXPHOS levels, potentially influenced by benign cells during metastasis. While the exact mechanisms behind OXPHOS deficiency in AD remain a scientific puzzle, recent evidence highlights the significance of mitochondrial F1Fo ATP synthase dysfunction in AD-related mitochondrial OXPHOS failure. Overall, mitochondrial heme, ER-mitochondria interactions, and calcium homeostasis are key players in the context of lung cancer progression, making them potential targets for therapeutic intervention. MAM alterations, Aβ production, and mitochondrial dysfunction play pivotal roles in AD pathogenesis. *C. elegans* models mimic AD features and are valuable for testing potential therapies. β and γ-secretases, found in MAMs, process Aβ protein precursor, contributing to Aβ production. Aβ accumulation at mitochondria-ER contact sites disrupts their function, a key step in AD neurodegeneration.

Various biological processes, including impaired cell proliferation and survival pathways, are significant contributors to this inverse relationship between AD and cancer. Additional studies should be carried out to establish the relationship. Shared biological mechanisms such as Pin1, Wnt, and p53 signaling play opposing roles in cancer and AD, leading to uncontrolled cell proliferation and survival in cancer and cell death and neurodegeneration in AD. The genes Pin1, p53, and Wnt are differentially regulated in these two conditions, with Pin1 serving a significant dual function. In cancer, Pin1 is often overexpressed and promotes oncogenic signaling pathways, while in AD, its loss contributes to neurodegeneration. Furthermore, in cancer, somatic p53 mutations are prevalent, while in AD, p53 activation is disrupted, impacting DNA repair and apoptosis. Similarly, the Wnt pathway plays a crucial role in both conditions, with upregulated Wnt signaling contributing to tumor development in cancer and downregulated Wnt signaling associated with Aβ neurotoxicity in AD. Moreover, mitochondrial interactions with Wnt, Pin1, and p53 pathways influence cancer and AD, warranting further investigation for tailored therapeutics and potential cancer treatments, particularly in p53-altered tumors. Future research may focus on understanding the molecular mechanisms of Pin1, Wnt, and p53 signaling in cancer and AD. Targeting these pathways therapeutically and exploring their dual roles may offer novel treatments for both conditions.

## Figures and Tables

**Figure 1 biology-13-00185-f001:**
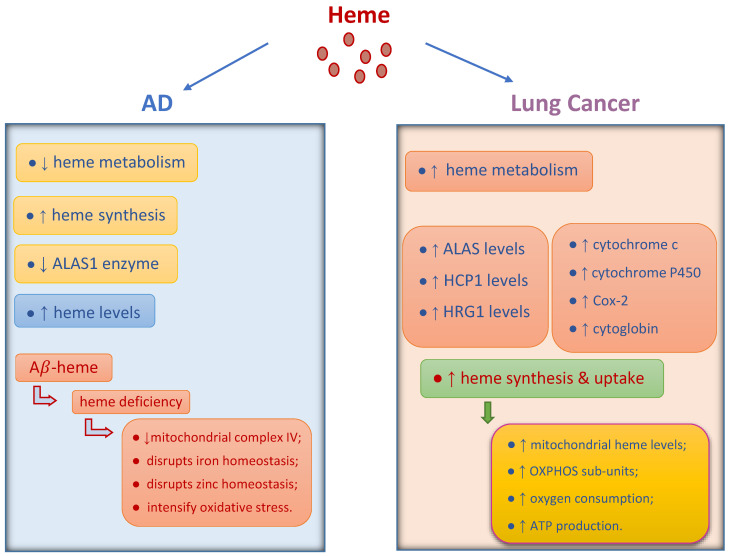
Heme’s effects in Lung Cancer and AD. In this figure, ↑ represents an increase, while ↓ indicates a decrease.

**Figure 2 biology-13-00185-f002:**
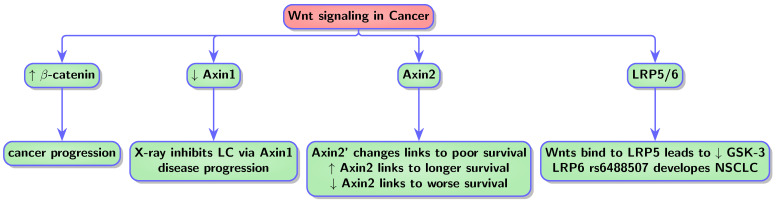
Wnt Signaling in Lung Cancer (LC). Dysregulated Wnt/β-catenin signaling leads to nuclear β-catenin accumulation (indicated by ↑), driving cancer progression. Axin1 downregulation (indicated by ↓) is common in LC, potentially reversible with X-ray treatment, impacting patient prognosis. High Axin2 expression (indicated by ↑) is linked to a longer survival period in LC, while low Axin2 (indicated by ↓) is associated with worse survival. LRP5 downregulates GSK-3 activity (indicated by ↓), while LRP6 rs6488507 polymorphism increases the risk of developing NSCLC.

**Figure 3 biology-13-00185-f003:**
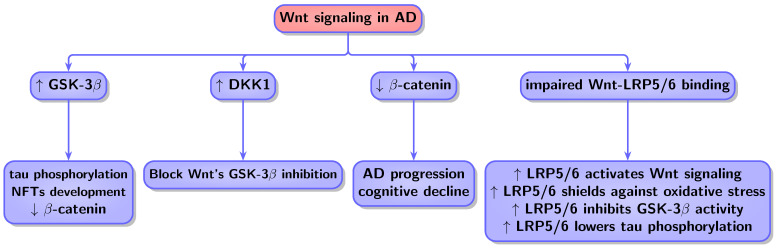
Wnt Signaling in AD. Increased activity of GSK-3β is indicated by ↑. A constitutively active GSK-3β leads to abnormal tau phosphorylation, the development of NFTs, and reduced levels of β-catenin (indicated by ↓). Increased DKK1 expression (indicated by ↑) in the AD brain hinders Wnt’s inhibition of GSK3-β. Overexpressing LRP5/6 (indicated by ↑) activates Wnt signaling, protects against oxidative stress, inhibits GSK-3β activity, and reduces tau phosphorylation.

**Figure 4 biology-13-00185-f004:**
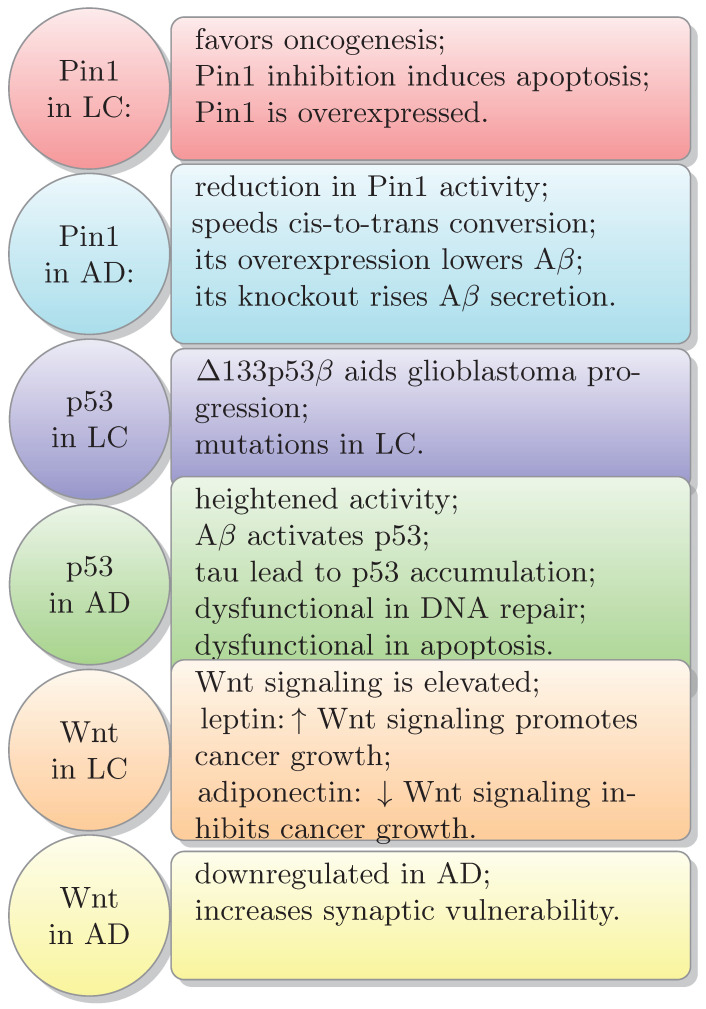
Common Signaling Pathways in Lung Cancer (LC) and AD.

**Table 1 biology-13-00185-t001:** Effects of Heme Oxygenase on AD and Cancer pathology.

	Effects	References
Cancer	shows a positive correlation with lymph node metastasis in NSCLC promotes tumor cell invasion and proliferation *HMOX1* in NSCLC: attenuates cell proliferation, modulates angiogenesis, attenuates metastasis. HSPs in NSCLC cells: decreases heme uptake, inhibits tumorigenic functions, slows the growth of NSCLC, decreases oxygen consumption rates, decreases ATP levels.	Degese et al., 2012, [86] Hsu et al., 2015, [87] Hsu et al., 2017, [89] Skrzypek et al., 2013, [95] Sohoni et al., 2019, [30]
AD	reduces pro-inflammatory cytokines: IL-1β, TNF-α, IL-6 maintenance of heme homeostasis protects against Aβ toxicity protects cells from iron-dependent toxicity long-term HO overexpression: induces tau phosphorylation, damages synaptic plasticity, abnormal iron buildup, impair mitochondrial function, reduces cognitive ability, induces ferroptosis. moderate duration and levels of HO: has anti-inflammatory effects, reduces oxidative stress, protects the blood–neural barrier.	Intagliata et al., 2019, [79] Chen et al., 2018, [115] Liu et al., 2020, [114] Hettiarachchi et al., 2014, [110] Hettiarachchi et al., 2017, [111] Choi et al., 2022, [39] Wang et al., 2015, [105] Hui et al., 2011, [106] Li et al., 2015, [107] Schipper et al., 2019, [113] Choi et al., 2022, [39]

## Data Availability

No new data were created or analyzed in this study. Data sharing is not applicable to this article.

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
