# Peer review of "Putative Molecular Mechanisms Underpinning the Inverse Roles of Mitochondrial Respiration and Heme Function in Lung Cancer and Alzheimer’s Disease"

_biology, 2024, doi:10.3390/biology13030185_

Round 1
Reviewer 1 Report (New Reviewer)
Comments and Suggestions for Authors
The present review summarizes/discusses roles of mitochondrial respiration and heme function in lung cancer and Alzheimer’s disease by citing the large numbers up to 340 references. The manuscript is well written and readable. I think that this interesting work would certainly advance our understanding of the roles of mitochondrial respiration and heme in those diseases. I strongly recommend this paper for publication in the Journal. However, I raise some concerns that need to be addressed before publication. If those concerns are adequately address in the revised manuscript, this interesting review would be significantly strengthened.
Concerns that need to be addressed.
[1] I would strongly suggest the authors to incorporate “List of abbreviations” before or after the main text. For example, please define ROS, AD, OXPHOS, T2D, Abeta, NSCLC, HO, SPP, CORM-1, SENSN2, APP, BACE1 and many others, if those appear in the main text, tables, figure legends, and figures more than twice.
[2] Fig. 1 is too small. It is difficult for general readers to grasp the points. Please significantly enlarge this figure.
[3] The title says “ …..Molecular Mechanism…”, but to my eyes, molecular mechanism is less described. Molecular mechanism should be more discussed by incorporating cartoons as figures, demonstrating the interactions/relationships between various functions described in this review. Please refer to cartoon figures used in the excellent review articles as mentioned in [4] and [5].
[4] Heme is emphasized in Abstract and Introduction. But the text does not discuss heme itself in detail, whereas it largely focuses HO. I would suggest the authors to discuss the interaction of heme or iron with p53 by citing some of the following or other relevant papers.
Cell Death Disease (2023) 14:710, J. Inorg. Biochem. (2023) 243, 112180, Mol. Cell (2023) 83, 1030, Nat. Commun. (2022) 13:7400, J. Biol. Inorg. Chem. (2022) 27, 393, Chem. Soc. Rev. (2019) 48, 5624, Cell Rep. (2014) 7, 180.
[5] HO produces free iron. Iron is associated with ROS generation, ferroptosis and other important physiological/pathological functions. Please discuss ferroptosis or iron functions more by citing some of the following or other relevant papers.
Mol. Cell (2023) 83, 1030, Age. Res. Rev. (2023) 87, 101899, Trend Cell Biol. (2023) 33 (12) 1062, Biomed. Pharmacother. (2023) 165, 114706, Biomed. Pharmacother. (2023) 159, 114241, Pharmacol. Therapeu. (2023) 244, 108373, Trend Pharmacol. Sci. (2023) 44 (10) 674, FEBS J. (2023) 290, 1688, Coord. Chem. Rev. (2023) 480, 215024, FEBS J. (2022) 289, 7810, Nat. Metab. (2023) 5, 10 & 111,
[6] Title and subtitles mention Cancer first and Alzheimer second in this order. But Table 1 describes opposite order, AD first and Cancer second. This order should be fixed.
[7] Table 1. It is difficult for readers to correspond each effect to each reference, all references bound together. Please fix this problem in order to be recognized which is which.
[8] References: Various forms are mixed in the present paper.
(1) Does the Journal policy suggest the authors to use abbreviated journal names, or non-abbreviated names? References of the present manuscript are mixed. For example, refs. 5, 13, 19, 87, 88, 101, 110, 111 and other references use the abbreviated form, whereas other references use non-abbreviated journal names. Please fix this according to the Journal policy.
(2) Also, description of the first and last pages of references is recommended or should only the first page be cited? For example, only first pages are described for refs. 2, 3, 5, 7, 9, 11 and others, whereas both first and last pages are described for other references. Please unify those according to the Journal policy.
Again, if those concerns are adequately addressed in the revised manuscript, the present excellent review would be largely improved. I strongly recommend publication of this superb review after revisions are made.
Author Response
Please find it in the attachment.

Reviewer 2 Report (New Reviewer)
Comments and Suggestions for Authors
See PDF file.

See PDF file.
Author Response
Please find it in the attachment.

Reviewer 3 Report (New Reviewer)
Comments and Suggestions for Authors
1) Please provide specific mechanisms through which adiponectin influences cancer and AD development, like adiponectin levels, its receptors and downstream signaling pathways.
2) The author need to elaborate on how oxidative stress, DNA damage, and inflammation contribute to development of COPD, lung cancer and AD.
3) Please provide brief explanation of the role of p53 in AD, on how p53 responds to stress conditions, its interaction with Aβ, and its involvement in apoptosis and DNA repair processes in the context of AD
4) Please provide potential future directions for research in understanding the roles of Pin1, Wnt, and p53 signaling in both cancer and AD.
5) The author need to provide more in-depth molecular insights, especially in relation to specific signaling pathways or molecular events that drive mitochondrial changes in cancer.
Comments on the Quality of English LanguageThe author need to work on grammatical errors.
Author Response
Please find it in the attachment.

Round 2
Reviewer 2 Report (New Reviewer)
Comments and Suggestions for Authors
None
This manuscript is a resubmission of an earlier submission. The following is a list of the peer review reports and author responses from that submission.
Round 1
Reviewer 1 Report
Comments and Suggestions for Authors
See attached file. I'm afraid my report would take far too long given the quality of the manuscript as it stands. I would suggest taking a look at my comments for the sections I have covered and applying them throughout the manuscript. The standard of proof reading is particularly low and the quality of grammar seems to drop off dramatically at page 8.

Seems to lack proof reading. There are many grammatical errors - too many to detail individually in a report.